# PROVABLE ROBUST WATERMARKING FOR AI-GENERATED TEXT

**Xuandong Zhao     Prabhanjan Ananth     Lei Li     Yu-Xiang Wang**
UC Santa Barbara
`{xuandongzhao,prabhanjan,leili,yuxiangw}@cs.ucsb.edu`

## ABSTRACT

We study the problem of watermarking large language models (LLMs) generated text — one of the most promising approaches for addressing the safety challenges of LLM usage. In this paper, we propose a rigorous theoretical framework to quantify the effectiveness and robustness of LLM watermarks. We propose a robust and high-quality watermark method, UNIGRAM-WATERMARK, by extending an existing approach with a simplified fixed grouping strategy. We prove that our watermark method enjoys guaranteed generation quality, correctness in watermark detection, and is robust against text editing and paraphrasing. Experiments on three varying LLMs and two datasets verify that our UNIGRAM-WATERMARK achieves superior detection accuracy and comparable generation quality in perplexity, thus promoting the responsible use of LLMs. Code is available at `https://github.com/XuandongZhao/Unigram-Watermark`.

## 1 INTRODUCTION

Generative Artificial Intelligence (AI) (Brown et al., 2020; Ramesh et al., 2022; Saharia et al., 2022; OpenAI, 2023a) has achieved significant progress in recent years, spanning from computer vision (CV) to natural language processing (NLP). Large language models (LLMs) such as ChatGPT (OpenAI, 2022) can generate coherent and contextually relevant long-form text in response to user-specified prompts. However, the ease of using LLMs has raised concerns about their potential misuse (Zellers et al., 2019; Weidinger et al., 2021; Stokel-Walker, 2022). For example, LLMs could be used to generate fake news, contaminate web content, or assist in academic dishonesty. Additionally, the proliferation of synthetic data from LLMs poses challenges for training new models, as synthetic data needs to be detected and excluded before model training (Radford et al., 2022; Carlini et al., 2023).

There are two main camps of existing attempts to address these challenges. One camp, inspired by Turing (1950), aims at generically distinguishing machine-generated text from that of the humans (Gehrmann et al., 2019; Mitchell et al., 2023; Hovy, 2016; Zellers et al., 2019; OpenAI, 2023b). These works primarily leverage hand-crafted or learned "statistical patterns" of generated text, thus their performance is not robust to distribution changes (e.g., by prompting / conditioning), prone to biases (Liang et al., 2023), and vulnerable to adversarial attacks.

The other camp advocates active intervention by injecting carefully-designed watermarks to machine-generated text (Kirchenbauer et al., 2023; Zhao et al., 2023). The watermarking approach does not search for statistical patterns (which could be hit-or-miss), but rather deliberately *plant* subtle but distinctive patterns within the content to enable downstream detection. Compared to the passive detection approaches, the watermarking methods aim at determining whether the text is coming from a *specific* language model rather than solving the Turing test generically. As a result, watermarking approaches are robust to distribution-shift and can essentially *prove* — rather than *predict* — the origin of the suspect text.

The most notable challenge for the watermarking approach is that the planted patterns could be post-processed away. As an example, Kirchenbauer et al. (2023)'s soft watermarking method divides the vocabulary into a "green list" and a "red list" based on the prefix token, and subtly increases the probability of choosing from the green list. If the watermarked sentence is edited by changing every other token into its synonym, then it is no longer possible to determine the green/red lists for each

candidate token, thus ruining the detector. One could also simply paraphrase the sentence as a whole using another off-the-shelf LLM.

In this paper, we take a first stab at formally defining robustness in the context of watermarking LLMs. Our contributions are fourfold.

1. We devise a rigorous theoretical framework for quantifying the performance drop, the correctness of detection, and the security property against post-processing.

2. We propose to simplify the scheme of Kirchenbauer et al. (2023) by using a fixed Green-Red split consistently and show that the new watermark, named UNIGRAM-WATERMARK, is *twice as robust* to edits as the baseline, provably.

3. We prove that the watermarked LLM is close to the original LLM (in all Renyi divergences) and show that the Type I/Type II errors of the detection algorithm decay exponentially as the suspect text length gets longer and more diverse.

4. We conduct experiments utilizing various large language models on diverse datasets. The results indicate that our method achieves superior detection accuracy and improved robustness against different attacks, thus promoting the responsible use of LLMs.

To the best of our knowledge, we are the first to obtain provably robust guarantees for watermarks for LLMs against arbitrary edits.

**Related work.** We build upon the work of Kirchenbauer et al. (2023) in which the family of $K$-gram (statistical) watermark was proposed[1]. The main method we consider chooses $K = 1$, thus its name UNIGRAM-WATERMARK. Our work provides formal theoretical guarantees to this family of $K$-gram watermark. For the sake of a clean presentation, we focus on the case when $K = 1$ and discuss the applicability of our results for $K > 1$ in the discussion section. Our work is independent of the concurrent work of cryptographic watermarks (Aaronson, 2023; Christ et al., 2023). In particular, Aaronson (2023)'s proprietary work can also be viewed as an alternative $K$-gram watermark, but uses a cryptographic approach for measuring utility drop, which results in a different kind of tradeoff. We defer detailed discussion to an extended discussion of the related work in Appendix A. Technically, the main theoretical tool we used for analyzing dependent random variables and their concentration tightly is due to Albert (2019), the instantiation to our problem is new and nontrivial.

## 2 PROBLEM SETUP AND METHOD

We start with an overview of the language model watermarking problem. The definitions and notations introduced in this section will be used throughout the paper.

**Language models.** A language model (LM) $\mathcal{M}$ is a statistical model that describes the probability of a sequence of words occurring in a sentence. Common neural language models (e.g., GPT-2/3 (Radford et al., 2019; Brown et al., 2020)) are designed for next-word prediction which typically uses a transformer neural network (Vaswani et al., 2017). The LM has a "vocabulary" $\mathcal{V}$ with $N := |\mathcal{V}| = 50,000$ tokens or more (Radford et al., 2019; Liu et al., 2019). Let $x$ be an input prompt. $y := [y_1, \ldots, y_n]$ are $n$ tokens generated by $\mathcal{M}$. During inference, $\mathcal{M}$ receives the input prompt $x$ as the prefix of generation. It iteratively computes logit scores $\ell_t$ for every next token. The logits transform into a probability distribution via soft-(arg)max function $\mathbf{p}_t[v] = \frac{\exp(\ell_t[v])}{\sum_{i \in \mathcal{V}} \exp(\ell_t[i])}$ for all $v \in \mathcal{V}$. The LM then samples the next token from this distribution: $y_t \sim \mathbf{p}_t$.

### 2.1 DEFINITION OF LANGUAGE MODEL WATERMARKING

In the language model watermarking problem, the objective for the model owner is to embed a secret message known as "watermark" within the generated sequence $y$ for a given prompt $x$. There are two desired requirements for watermarking. First, the quality of the watermarked model should be comparable to the quality of the original, un-watermarked model. Second, an adversary needs to modify sufficiently many AI-generated text in order to evade detection.

---

[1]Note the changed name. Kirchenbauer et al. (2023) referred to its (unnamed) soft-watermark that determines the green/red list using a prefix of length $(K - 1)$. We think $K$-gram watermark is the most concise and informative name for this family.

**Definition 2.1** (Edit distance). The edit distance, denoted as $\mathsf{ED}(\boldsymbol{y}, \boldsymbol{z})$, quantifies the number of basic operations required to transform a sequence $\boldsymbol{y}$ into another sequence $\boldsymbol{z}$. These operations include "insertion", "deletion", and "replacement" of tokens.

**Definition 2.2** (Language model watermarking). A **language model watermarking scheme** consists of two probabilistic polynomial-time algorithms (Watermark, Detect):

- Watermark($\mathcal{M}$): Let $\mathcal{M}$ be a language model and let $\mathbf{p}_t := \mathbb{P}_{\mathcal{M}(\boldsymbol{x})}[y_t = \cdot | \boldsymbol{y}_{1:t-1}]$ be the *conditional* probability distribution of $t$-th token on $\mathcal{V}$ generated by $\mathcal{M}$. This algorithm produces a new model $\hat{\mathcal{M}}$ with a new conditional distribution $\hat{\mathbf{p}}_t := \mathbb{P}_{\hat{\mathcal{M}}(\boldsymbol{x})}[y_t = \cdot | \boldsymbol{y}_{1:t-1}]$ on $\mathcal{V}$. Additionally, it outputs a detection key k associated with $\hat{\mathcal{M}}$. The watermark could contain certain randomness.

- Detect(k, $\boldsymbol{y}$): This algorithm takes input detection key k and sequence $\boldsymbol{y}$, then outputs 1 (indicating it was generated by $\hat{\mathcal{M}}$) or 0 (indicating it was *not* generated by $\hat{\mathcal{M}}$).

We require the following **three correctness properties** to hold:

- $\omega$-Quality of watermarked output, for $\omega \in \mathbb{R}$: Assume the original language model $\mathcal{M}$ generates a probability vector $\mathbf{p}_t$ for the token at position $t$. The watermarked model $\hat{\mathcal{M}}$ predicts the token at position $t$ using the modified probability vector $\hat{\mathbf{p}}_t$. It is required that the distance between the two probability distributions satisfies: $D\left(\hat{\mathbf{p}}_t \| \mathbf{p}_t\right) \leq \omega$ for any fixed prompts and prefixes.

- $\alpha_{\boldsymbol{y}}$-Type I error ("No false positives"): for any fixed $\boldsymbol{y}$ (i.e., independent to k), it holds that $\mathbb{P}\left[\mathsf{Detect}(\mathsf{k}, \boldsymbol{y}) = 1 \; ; \; (\hat{\mathcal{M}}, \mathsf{k}) \sim \mathsf{Watermark}(\mathcal{M})\right] \leq \alpha_{\boldsymbol{y}}$.

- $\beta_{(\boldsymbol{x}, \mathcal{M})}$-Type II error ("No false negatives"): $\mathbb{P}\left[\mathsf{Detect}(\mathsf{k}, \boldsymbol{y}) = 0 \; ; \; \begin{smallmatrix}(\hat{\mathcal{M}}, \mathsf{k}) \sim \mathsf{Watermark}(\mathcal{M}) \\ \boldsymbol{y} \sim \hat{\mathcal{M}}(\boldsymbol{x})\end{smallmatrix}\right] \leq \beta_{(\boldsymbol{x}, \mathcal{M})}$.

We also require the following **security property** (parameterized by $\epsilon \geq 0$ and $\eta(\boldsymbol{y}, \mathsf{k}, \epsilon)$):

- For any adversary $\mathcal{A}$ that postprocesses $\boldsymbol{y}$ with auxiliary information aux and any prompt $\boldsymbol{x} \in \mathcal{V}^*$

$$\mathbb{P}\left[\mathsf{Detect}(\mathsf{k}, \boldsymbol{y}_{\mathcal{A}}) = 1 \text{ or } \mathsf{ED}(\boldsymbol{y}, \boldsymbol{y}_{\mathcal{A}}) \geq \eta(\mathsf{k}, \boldsymbol{y}, \epsilon) \Big|_{\mathsf{Detect}(\mathsf{k},\boldsymbol{y})=1}^{\boldsymbol{y},\mathsf{k},} \; ; \; \begin{smallmatrix}(\hat{\mathcal{M}}, \mathsf{k}) \sim \mathsf{Watermark}(\mathcal{M}) \\ \boldsymbol{y} \sim \hat{\mathcal{M}}(\boldsymbol{x}) \\ \boldsymbol{y}_{\mathcal{A}} \sim \mathcal{A}(\boldsymbol{y}, \mathsf{aux})\end{smallmatrix}\right] \geq 1 - \epsilon.$$

*Remark* 2.3 (Discussion on Definition 2.2). Informally, our definition allows us to formally quantify the essential properties of a language model watermarking scheme including its generation quality relative to the input LM, the accuracy of detection in terms of both false positives and false negatives, as well as the robustness to attacks.

The security property, in particular, states the following: suppose a malicious adversary intends to evade the detection algorithm, then the adversarial answer, to some input prompt $\boldsymbol{x}$, should be far away (in edit distance) from any AI-generated answer. In other words, the optimal strategy to evade the detection algorithm would necessitate executing a minimum number of insert/delete/replacement operations, captured by the function $\eta(\cdot)$ in Definition 2.2. This conceptually suggests that the adversary must exert considerable effort to successfully elude detection.

Admittedly, there are other attacks where edit distance does not capture either the effort or the utility loss. For example, if one prompts an unwatermarked LLM to paraphrase $\boldsymbol{y}$ then the number of edits can be large but the semantic meaning is retained. However, edit distance is a natural metric that smoothly interpolates the gray zone between the world where $\boldsymbol{y}_{\mathcal{A}} = \boldsymbol{y}$ in which it should clearly be caught and the other world where $\boldsymbol{y}_{\mathcal{A}}$ is *independently created* without using $\hat{\mathcal{M}}$ in which it would be a false positive if Detect returns 1.

## 2.2 THREAT MODELS

**Adversary's objective.** The primary objective of the adversary is to render the watermark detection algorithm ineffective. Specifically, the adversary aims to produce a $\boldsymbol{y}_{\mathcal{A}}$ such that $\mathsf{Detect}(\mathsf{k}, \boldsymbol{y}_{\mathcal{A}}) = 0$ while at the same time, $\boldsymbol{y}_{\mathcal{A}}$ is a minor modification of an AI-generated text $\boldsymbol{y}$.

**Adversary's capabilities.** We consider an adversary with black-box input-output access to the language model. This adversary has the capacity to modify the sequence within a *bounded edit*

*distance*. Given an input prompt $\boldsymbol{x}$, the watermarked language model generates a text output $\boldsymbol{y} \leftarrow \hat{\mathcal{M}}(\boldsymbol{x})$. The adversary, equipped with arbitrary side-information and computational resources, can then produce a modified output $\boldsymbol{y}_{\mathcal{A}}$ such that the edit distance between the original and modified output, $\mathsf{ED}(\boldsymbol{y}, \boldsymbol{y}_{\mathcal{A}})$, is bounded, i.e. $\mathsf{ED}(\boldsymbol{y}, \boldsymbol{y}_{\mathcal{A}}) < \eta$.

## 2.3 METHOD

---
**Algorithm 1** UNIGRAM-WATERMARK: Watermark
---
1: **Input:** random number generator $F$, green list size $\gamma \in (0, 1)$, watermark strength $\delta$.
2: Randomly generate a watermark key k using $F$.
3: Use watermark key to partition the vocabulary of $\mathcal{M}$ into a "green list" $G \subset \mathcal{V}$ of size $\gamma|\mathcal{V}|$, and a "red list" $R = G^c$.
4: Define a new language model $\hat{\mathcal{M}}$ where for $t$ and any prefix $[\boldsymbol{x}, \boldsymbol{y}_{1:t-1}]$, the resulting logits satisfy
$$\hat{\boldsymbol{\ell}}_t[v] := \boldsymbol{\ell}_t[v] + \delta \mathbf{1}(v \in G), \tag{1}$$
where $\mathbf{1}(\cdot)$ is the indicator function and the logit vector $\boldsymbol{\ell}_t \in \mathbb{R}^{|\mathcal{V}|}$ is obtained by the passing the same prefix to $\mathcal{M}$.
5: **Output:** watermark key k, watermarked language model $\hat{\mathcal{M}}$.
---

---
**Algorithm 2** UNIGRAM-WATERMARK: Detect
---
1: **Input:** suspect text $\boldsymbol{y}$, watermark detection key k, threshold $\tau$.
2: **Output:** 1 or 0 (whether the text is watermarked).
3: Use the watermark detection key k to find the "green list" $G$.
4: Calculate the number of green list tokens $|\boldsymbol{y}|_G = \sum_{t=1}^n \mathbf{1}(y_t \in G)$ in $[y_1, \ldots, y_n]$.
5: Compute the $z$-statistic:
$$z_{\boldsymbol{y}} = \left(|\boldsymbol{y}|_G - \gamma n\right) / \sqrt{n\gamma(1-\gamma)}. \tag{2}$$
6: **if** $z_{\boldsymbol{y}} > \tau$ **then return** 1, i.e., "The suspect text is watermarked."
7: **else return** 0, i.e., "The suspect text is not watermarked."
---

Now let us instantiate Definition 2.2 with concrete algorithms. We will focus on UNIGRAM-WATERMARK — a variant of the $K$-gram watermark proposed by Kirchenbauer et al. (2023) but with a choice of $K = 1$. Pseudocodes of our approach Watermark and Detect are provided in Algorithm 1 and 2. In Algorithm 1, we randomly partition the vocabulary into two distinct sets: the green list with $\gamma N$ tokens and the red list with the remaining tokens. In $\hat{\mathcal{M}}$, the logits of the language model for the green list tokens are increased by $\delta$ while the logits for tokens in the red list remain unchanged. Then at detection time (Algorithm 2), we count the number of green tokens in the suspect text, normalize the test-statistic, then make a calibrated decision on whether we think the suspect text is generated from $\hat{\mathcal{M}}$ or not. We show the examples of real prompts and watermarked outputs in Table 1.

The watermarking procedure is parameterized by two *watermark strength parameters* $\gamma, \delta$. $\gamma$ determines the fraction of the vocabulary included in the green list. We typically set $\gamma$ to be a constant, e.g., $1/3$ or $0.5$. $\delta$ specifies the increase in the logits associated with the green list tokens. The larger $\delta$ is, the lower the quality of the watermarked LM, but the easier it is to detect.

Our UNIGRAM-WATERMARK enjoys all good properties of the general $K$-gram watermark from Kirchenbauer et al. (2023). It runs in linear time and does not require access to the language model or the prompt used for generation. It is also intuitively robust to cropping and minor edits.

Overall, the proposed watermarking scheme requires almost no overhead in its implementation, is extremely simple, and is easy to maintain. The big question is: `How well does this watermark scheme work?`

The remainder of this paper provides answers to this question with provable guarantees (Section 3) on the properties from Definition 2.2 and extensive experiments (Section 4).

Before that, let us address two burning questions that a knowledgeable reader may have.

| Prompt | Is it possible to fail several times and succeed once? |
|---|---|
| **LLaMA-13B** (un-watermarked) PPL: 12.19 $z$-score: -2.425 | Of course it is, and that is how we improve.\nSaying "I can't do that" is never a good thing.\nSometimes we think we've tried all we can and that "isn't enough". That is the time when we ask for help.\nThe root of all evils is to be a secret. Honesty and self-criticism is necessary for improvement. \nThe measure of intelligence is the ability to change. [continues...] |
| **LLaMA-13B** (watermarked) PPL: 12.47 $z$-score: 11.085 | When most people are confronted with failure, they cannot imagine such a thing happening. When one faces business reverses and bankruptcy, it seems impossible. When we are rejected it looks as if we are going to be rejected forever. However, it does not need to be this way. The human spirit simply will not give up. When we face failure and it looks like we [continues...] |

Table 1: Comparison of un-watermarked and watermarked text using the LLaMA-13B model. Green and red tokens are color-coded respectively. UNIGRAM-WATERMARK produces watermarked text of similar quality without noticeable degradation, yet with significant differences in $z$-scores for watermark detection.

**Why choosing $K = 1$?** Recall that the general $K$-gram watermark works in the same way as ours, but randomly generates a different Green list for each prefix of length $K - 1$. In contrast, choosing $K = 1$ means we have a consistent green list for every new token the language model generates. The main advantage of choosing $K = 1$ is that it is the most robust choice within this family — and we believe robustness is the single most important feature of a watermarking scheme in practice.

**Robustness to other attacks.** Besides the robustness to edits, which we will prove in Section 3 and compare to that of $K \geq 2$. UNIGRAM-WATERMARK is also resilient to many other kinds of generation time attacks that people can apply such as reversing, shuffling, as well as the "Emoji insertion attack" that will completely break the watermark for $K \geq 2$ but not for $K = 1$. We provide a detailed discussion of this in Appendix E.1.

**The price for robustness?** Kirchenbauer et al. (2023) did not consider the choice of $K = 1$ for an obvious reason. The watermark is now so simple that an attacker who observes the generated text may learn to guess the consistent green list. This is an issue for $K \geq 2$ too but certainly more so for $K = 1$. There is a robustness-learnability tradeoff as we adjust $K$ which deserves a more rigorous treatment in future work. That said, we are ready to argue for biasing towards robustness. Why? We argue that in practice, it could be surprisingly difficult for an attacker to construct a meaningful attack when they do not have access to the original LM. We provide a more detailed experimental study with a faithful practical attack in Appendix B.4. Moreover, there are alternative ways to get around this issue by refreshing the green list once in a while.

## 3 MAIN THEORETICAL RESULTS

In this section, we present the quality, correctness, and security properties of UNIGRAM-WATERMARK as described in Definition 2.2.

### 3.1 QUALITY GUARANTEE OF UNIGRAM-WATERMARK

We first show that the distance between the original probability vector $\mathbf{p}_t$ and the watermarked probability vector $\hat{\mathbf{p}}_t$ are very close to each other in any Renyi-divergence.

**Theorem 3.1.** *Consider $\boldsymbol{h}$ as the input to the language model at step $t$, denoted as $\boldsymbol{h} = [\boldsymbol{x}, \boldsymbol{y}_{1:t-1}]$. Fix green list $G$. Let $\delta$ represent the watermark strength. For any $\boldsymbol{h}$, the $\alpha$-th order Renyi-divergence between the watermarked probability distribution $\hat{\mathbf{p}}_t = \hat{\mathbf{p}}_t(\cdot|\boldsymbol{h})$ at time step $t$ and the original probability distribution $\mathbf{p}_t = \mathbf{p}_t(\cdot|\boldsymbol{h})$ satisfies:*

$$\forall \boldsymbol{h}, \max\left(D_\alpha\left(\hat{\mathbf{p}}_t\|\mathbf{p}_t\right), D_\alpha\left(\mathbf{p}_t\|\hat{\mathbf{p}}_t\right)\right) \leq \min\{\delta, \alpha\delta^2/8\}.$$

The proof, deferred to the appendix, leverages a surprising connection to modern techniques in the differential privacy literature (Dwork et al., 2006; Dong et al., 2020).

*Remark* 3.2 (KL-divergence and other probability distance metrics). Renyi-divergence is very general. Kullback-Leibler-divergence and chi-square divergence are directly implied by the $\alpha$-Renyi divergence bound of $\min\{\delta, \alpha\delta^2/8\}$ by choosing $\alpha = 1$ and $\alpha = 2$ respectively and swap $\hat{\mathbf{p}}$ and $\mathbf{p}$. Hellinger distance can be obtained by choosing $\alpha = 0.5$. By Pinsker's inequality, we get a Total

Variation distance bound of $\min\{\sqrt{\delta/2}, \delta/4\}$. Moreover, by choosing $\alpha \to \infty$, we obtain an upper bound of $\delta$ for a very strong multiplicative guarantee known as max-divergence. The resulting two distributions $\hat{\mathbf{p}}$ and $\mathbf{p}$ are referred to by cryptographers as $(\delta, 0)$-*indistinguishable*, which says that for any measurable event $S$, the log-odds ratio satisfies $-\delta \le \log \frac{\hat{\mathbf{p}}_t(y_t \in S|\mathbf{h})}{\mathbf{p}_t(y_t \in S|\mathbf{h})} \le \delta$.

To summarize, our result shows that Algorithm 1 produces $\hat{\mathcal{M}}$ that satisfies $\omega$-quality of watermarked output with $\omega$ (as a function of $\delta$) for almost all commonly used probability distance $D$.

## 3.2 TYPE I ERROR OF UNIGRAM-WATERMARK

**Theorem 3.3** (No false positives (short version of Theorem C.4)). *Consider $\mathbf{y} = \mathbf{y}_{1:n}$ as any* fixed *text. Define $C_{\max}(\mathbf{y}) := \max_{i \in [N]} \sum_{j=1}^{n} \mathbf{1}(y_j = i)$ and $V(\mathbf{y}) := \frac{1}{n}\sum_{i=1}^{N}(\sum_{j=1}^{n}\mathbf{1}(y_j = i))^2$. With probability $1 - \alpha$ (over only the randomness of $G$):*

$$z_{\mathbf{y}} \le \sqrt{\frac{64V(\mathbf{y})\log(9/\alpha)}{1-\gamma}} + \frac{16C_{\max}(\mathbf{y})\log(9/\alpha)}{\sqrt{n\gamma(1-\gamma)}}.$$

The theorem says that the $z$-score for any sufficiently diverse text is $\tilde{O}(1)$ and it is applicable to any text not generated by the watermarked LM $\hat{\mathcal{M}}$.

*Remark* 3.4 (Controlling false positive rate). The theorem implies that if we choose $\tau > \sqrt{\frac{64V\log(9/\alpha)}{1-\gamma}} + \frac{16C_{\max}\log(9/\alpha)}{\sqrt{n\gamma(1-\gamma)}}$, then the false-positive rate is smaller than $\alpha$. Note that $V$ and $C_{\max}$ can be computed directly from $\mathbf{y}$, allowing us to choose an input-dependent $\tau$ as a function of $V, C_{\max}$ that achieves a $\alpha$-Type I error guarantee with a fixed $\alpha$ for all inputs. In particular, the Type I error $\alpha$ decreases exponentially as we increase the threshold $\tau$.

## 3.3 TYPE II ERROR OF UNIGRAM-WATERMARK

To bound the Type II error, i.e., false negative rates, we need to make certain assumptions about $\mathbf{p}$ of the language model and the prompt $\mathbf{x}$. These assumptions include a **"on-average high entropy"** assumption and a **"homophily"** condition. We will provide a detailed definition and discussion of these assumptions in Appendix C.4.1 and Appendix C.4.2.

The "on-average high entropy" assumption requires the probability of the roll-out text to be "sufficiently diverse" on average. It is related but different from the "spike entropy" assumption used by Kirchenbauer et al. (2023). The "homophily" assumption is new to this paper. It is an assumption about the distribution induced by the state-transitions of the language model $\mathcal{M}$, which says that increasing the probability of a green-list token at time $t$ does not decrease the probability of seeing that token in the future. This may seem counter-intuitive, but we will give concrete examples in Appendix C.4.2 to show why this is fundamental for any statistical watermark to work effectively.

**Theorem 3.5** (Only true positive (informal version of Theorem C.13)). *Assume "average-high entropy" and "homophilly" to be valid with appropriate parameters, and in addition $n \ge \tilde{\Omega}(\log(1/\beta)/\delta^2)$, then with probability $1 - \beta$,*

$$z_{\mathbf{y}} \ge \Omega\left((e^\delta - 1)\sqrt{n\gamma(1-\gamma)}\right).$$

*Remark* 3.6. The bounds on Type I/II error together say that $z_{\mathbf{y}} \asymp \delta\sqrt{n}$ if $\mathbf{y}$ is from $\hat{\mathcal{M}}$ while $z_{\mathbf{y}} \asymp O(1)$ otherwise, i.e., there is a large margin between them so we can choose $\tau$ in between. Also, the $\alpha$ and $\beta$ parameters decay exponentially as the $n$ gets larger.

## 3.4 SECURITY PROPERTY OF UNIGRAM-WATERMARK

We demonstrate the robustness of our watermarking scheme against editing attempts through Theorem 3.7. As a baseline of comparison, we also obtain new robustness guarantees for the soft watermarking method proposed in Kirchenbauer et al. (2023). The detailed proof is deferred to the Appendix C.

**Theorem 3.7** (Robustness to editing). *Let $\mathbf{y} = [y_1, \ldots, y_n]$ represent the watermarked sequence. Suppose the adversary $\mathcal{A}$ follows Definition 2.2 and outputs a modified text $\mathbf{u} = [u_1, \ldots, u_m]$.*

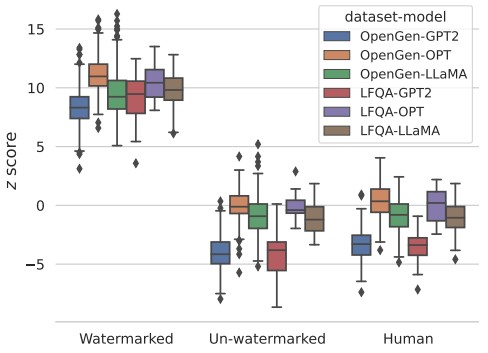
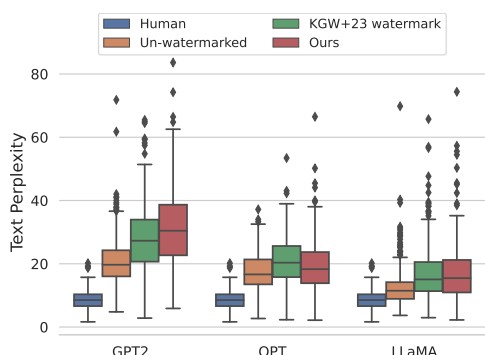

(a) $z$-scores of watermarked and un-watermarked machine-generated text, along with the $z$-score of human-generated text. The watermarked text $z$-score surpasses the empirical threshold of $z = 6.0$.

(b) Text perplexity comparison (evaluated by GPT-3) between human-generated text and text generated by various models on the OpenGen dataset.

Figure 1: $z$-score comparison and text perplexity comparison.

*Following Equation 2, we calculate $z$-score $z_{\boldsymbol{y}}$ and $z_{\boldsymbol{u}}$. Assume edit distance between $\boldsymbol{y}$ and $\boldsymbol{u}$ (denoted as $\eta$) satisfies $\eta < n$. Then we have*

$$z_{\boldsymbol{u}} \geq z_{\boldsymbol{y}} - \max\left\{\frac{(1+\gamma/2)\eta}{\sqrt{n}}, \frac{(1-\gamma/2)\eta}{\sqrt{n-\eta}}\right\}.$$

*In particular, when $\eta \leq \frac{2\gamma n}{(1+\gamma/2)^2}$, we can drop the second term in the max.*

This theorem bounds the changes to our test $z$-score when $\eta$ edits are performed. As we established for a high-entropy sequence, $z_{\boldsymbol{y}}$ typically grows in $O((e^\delta - 1)\sqrt{n})$, which means that when $\delta$ is a constant, with an appropriate choice of $\tau$, the watermark is robust up to $O(n)$ arbitrary edits! Finally, compared to Kirchenbauer et al. (2023)'s watermark, ours is twice as robust (see Appendix D).

## 4  EXPERIMENT

In this section, we aim to conduct experiments to evaluate watermark detection performance, watermarked text quality, and robustness against attacks compared to the baseline. Additional experiment results including different parameters, white-box attacks, scaled language models, etc. are deferred to Appendix B.

### 4.1  EXPERIMENT SETTING

**Datasets and prompts.** We utilize two long-form text datasets: OpenGen and LFQA. OpenGen, collected by Krishna et al. (2023), consists of 3K two-sentence chunks sampled from the validation split of WikiText-103 (Merity et al., 2017). The subsequent 300 tokens serve as the human-written continuation. LFQA is a long-form question-answering dataset created by Krishna et al. (2023) by scraping questions from Reddit, posted between July and December 2021, across six domains. Krishna et al. (2023) randomly select 500 questions from each domain and pair them with their corresponding longest human-written answers, resulting in 3K QA pairs. In our experiments, we use the questions as prompts and the corresponding answers as human-written text.

**Language models.** We conduct experiments using three state-of-the-art public language models of varying sizes from different model families: GPT2-XL with 1.5B parameters (Radford et al., 2019), OPT-1.3B (Zhang et al., 2022), and LLaMA-7B (Touvron et al., 2023). Nucleus Sampling (Holtzman et al., 2020) is employed as the default decoding algorithm to introduce randomness while maintaining human-like text output. The models are loaded from the Huggingface library (Wolf et al., 2019), and the `generate` API function is used to adjust the logits distribution of the language model.

**Evaluation methods.** Maintaining a low false positive rate is crucial to prevent misclassifying un-watermarked text as watermarked. To ensure this, we set the false positive rates at 1% and 10% for

| Setting | Method | OpenGen | | | | LFQA | | | |
|---|---|---|---|---|---|---|---|---|---|
| | | 1% FPR | | 10% FPR | | 1% FPR | | 10% FPR | |
| | | TPR | F1 | TPR | F1 | TPR | F1 | TPR | F1 |
| No attack | KGW+23 | 1.000 | 0.995 | 1.000 | 0.952 | 1.000 | 0.995 | 1.000 | 0.952 |
| | UNIGRAM-WATERMARK | 1.000 | 0.995 | 1.000 | 0.952 | 1.000 | 0.995 | 1.000 | 0.952 |
| ChatGPT | KGW+23 | 0.565 | 0.704 | 0.853 | 0.747 | 0.327 | 0.453 | 0.673 | 0.490 |
| | UNIGRAM-WATERMARK | 0.866 | 0.910 | 0.961 | 0.818 | 0.442 | 0.568 | 0.865 | 0.584 |
| DIPPER-1 | KGW+23 | 0.386 | 0.546 | 0.738 | 0.720 | 0.372 | 0.534 | 0.740 | 0.767 |
| | UNIGRAM-WATERMARK | 0.729 | 0.830 | 0.922 | 0.837 | 0.639 | 0.770 | 0.909 | 0.865 |
| DIPPER-2 | KGW+23 | 0.490 | 0.646 | 0.810 | 0.769 | 0.432 | 0.595 | 0.845 | 0.839 |
| | UNIGRAM-WATERMARK | 0.777 | 0.862 | 0.941 | 0.852 | 0.693 | 0.810 | 0.948 | 0.894 |
| BART | KGW+23 | 0.342 | 0.505 | 0.667 | 0.759 | 0.457 | 0.617 | 0.783 | 0.836 |
| | UNIGRAM-WATERMARK | 0.590 | 0.730 | 0.861 | 0.857 | 0.656 | 0.784 | 0.885 | 0.897 |

Table 2: Performance comparison of our method (UNIGRAM-WATERMARK) and the soft watermarking method proposed in Kirchenbauer et al. (2023) (denoted as KGW+23). Both methods employ LLaMA-7B with nucleus sampling, utilizing $\delta = 2.0$ and $\gamma = 0.5$. We use ChatGPT, DIPPER, and BART for paraphrasing the watermarked text as paraphrasing attacks. True positive rate and F1 score are presented for fixing the false positive rates at 1% and 10%. When there is no attack, both methods exhibit perfect watermark detection. Nevertheless, when subjected to paraphrasing attacks, UNIGRAM-WATERMARK consistently outperforms KGW+23.

all detection algorithms and adjust the detection threshold accordingly. We report true positive rate (TPR), F1 score, and ROC curves. GPT3 (`text-davinci-003`) (Ouyang et al., 2022), is used as the oracle model for perplexity evaluation. The experiments are conducted on Nvidia A100 GPUs.

## 4.2 WATERMARKING RESULTS

We use a watermark strength of $\delta = 2.0$ and a green list ratio of $\gamma = 0.5$. We also use different watermark keys k for different models. Stronger watermarks can be achieved for shorter sequences for a smaller $\gamma$ and a larger $\delta$. From the two datasets, we generate 500 watermarked sentences and 500 un-watermarked sentences using three different models (GPT2-XL, OPT-1.3B, and LLaMA-7B). We label them as "watermarked" and "un-watermarked" respectively. We also have corresponding human-written text for each prompt, referred to as "human". All sentences are cropped to a length of 200 tokens. $z$-scores are calculated for hypothesis testing as shown in Algorithm 2 between different sentence groups. The results (Figure 1a) indicate a clear distinction between watermarked and non-watermarked text. A default threshold of $z$-score = 6.0 can be used to determine if a text is watermarked. For a fair comparison with Kirchenbauer et al. (2023), we also set $\delta = 2.0$ and $\gamma = 0.5$ for their method.

Figure 1b demonstrates the text perplexity of human, un-watermarked machine-generated, and two watermarking-generated texts, evaluated on the OpenGen dataset. The perplexity of human text is significantly lower, likely due to the expertise contributed in the Wikipedia-based dataset used to train GPT3. We observe that the perplexity of the watermarked text is comparable to that of human-generated text, especially with the use of the largest model LLaMA-7B. This finding further supports the effectiveness of our method in preserving linguistic characteristics and coherence, ensuring seamless integration of watermarks without compromising overall text quality. One example of the prompt questions and machine-generated answers can be found in Table 1. We also conduct human evaluations to assess text quality. We enlist crowd workers from Amazon Mechanical Turk (AMT) to evaluate the quality of both watermarked and unwatermarked texts. From the LLaMA-7B model on the OpenGen dataset, we select 100 watermarked and 100 unwatermarked texts, anonymize the sentences, and ask workers to rate the quality on a scale of 1 (poor) to 5 (excellent). Each sentence undergoes two evaluations. The average score and standard deviation are computed and presented in Table 3.

| | Avg Score | STD |
|---|---|---|
| Un-watermarked | 3.660 | 0.655 |
| Watermarked | 3.665 | 0.619 |

Table 3: Human evaluation result.

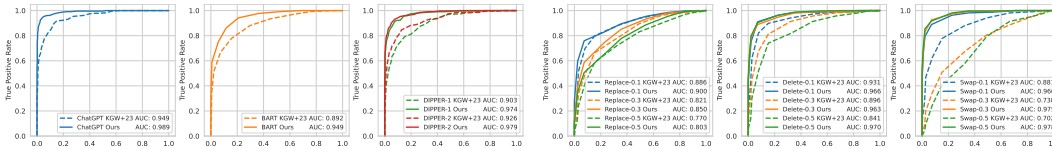

(a) UNIGRAM-WATERMARK against paraphrasing attacks on OpenGen dataset with LLaMA-7B.

(b) UNIGRAM-WATERMARK against editing attacks on LFQA dataset with LLaMA-7B.

Figure 2: ROC curves with corresponding AUC values for watermark detection against various attack methods. Complete results can be found in the Appendix B.

## 4.3 ROBUSTNESS RESULTS

**Paraphrasing attack.** To demonstrate the superior robustness of our method, supported by our theorem, we devise experiments to compare its performance against Kirchenbauer et al. (2023). We employ different paraphrase attack techniques targeting the removal of the watermark. Firstly, we utilized two versions of the DIPPER model (Krishna et al., 2023), we denote them as "DIPPER-1" and "DIPPER-2". DIPPER-2 has greater diversity than DIPPER-1. Additionally, we leverage the ChatGPT API, generating paraphrased text by providing prompts such as "*Rewrite the following paragraph:*". Furthermore, we employ BART (Lewis et al., 2019) (`bart-large-cnn`, a large-sized model fine-tuned on the CNN Daily Mail dataset (Hermann et al., 2015)) for text summarization as another type of paraphrasing attack. The results of our experiments are shown in Figure 2 and Table 2. The results illustrate the substantial improvement in robustness achieved by our method compared to Kirchenbauer et al. (2023). Notably, our method achieves an accuracy rate of over 85% with a false positive rate of 10%.

**Editing attack.** To further evaluate the robustness of UNIGRAM-WATERMARK against edit attacks, we examine its performance when subjected to synonym replacement, random deletion, and random swapping. These edit attack scenarios represent common techniques used to manipulate text and potentially remove watermarks. We conduct these attacks for the watermarked text of UNIGRAM-WATERMARK and KGW+23. The results are shown in Figure 2. In each scenario, our method consistently outperforms Kirchenbauer et al. (2023) watermarking scheme, showcasing its enhanced resilience and effectiveness in protecting the integrity of the embedded watermarks.

## 4.4 DISTINGUISHING HUMAN-WRITTEN TEXT

An interesting observation emphasized by Liang et al. (2023) is the misclassification of non-native English writing samples as AI-generated by existing AI content detectors. Our method can effectively establish text origin and maintain robustness to distribution shifts. We evaluate UNIGRAM-WATERMARK in distinguishing human-written text on a dataset of human-written TOEFL essays collected by Liang et al. (2023). Our method demonstrates a remarkable ability to accurately classify human-written text, as evidenced by significantly lower $z$-scores compared to the empirical threshold of $z = 6.0$. This outcome underscores the effectiveness of our watermark

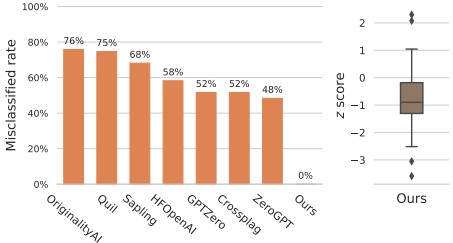

Figure 3: Distinguishing human-written text on TOEFL dataset.

in discerning text generated by human authors, further enhancing its practical utility and reliability.

## 5 CONCLUSION

In this paper, we have addressed the concerns surrounding the potential misuse of large language models and proposed an effective watermarking approach, UNIGRAM-WATERMARK, for detecting machine-generated text from a *specific* language model. Our contributions include the development of a rigorous theoretical framework, designing a provable effective, and robust watermarking scheme under this framework, as well as conducting extensive experiments to demonstrate the effectiveness and robustness of our method in practice. We anticipate that our work will inspire future research to develop more resilient watermarking methods capable of withstanding a broader range of attacks.

## 6 IMPACT STATEMENTS

**Applicability to general $K$-Gram watermark.** While we focused on UNIGRAM-WATERMARK, most of our results apply to $K$-Gram watermarks with $K \geq 2$ too. These include the Type I error bound, security properties (Robustness to edits), as well as the "Unique" alternative detector which we presented in Appendix E. While our Type II error bound does not *directly* work for $K \geq 2$, some of our intermediate steps can be applied.

**Limitations.** While our watermarking method, UNIGRAM-WATERMARK, demonstrates improved robustness against edits, its reliance on a fixed Green-Red split may not be universally optimal. The performance and robustness of watermarking methods can vary depending on the specific characteristics of the LLM and the generated text. Additionally, although our method enhances detection capabilities, it is not immune to all possible attacks.

**Future work.** Future work includes constructing *unlearnable* watermarks, understanding the robustness-learnability tradeoff as well as unifying cryptographical and statistical watermarks.

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

CONTENTS

## A  MORE ON RELATED WORK

**Watermarking natural languages.** The concept of watermarking, which involves hiding identifying information within data, has a long history. However, watermarking digital text has been challenging due to its discrete nature (Stefan et al., 2000). Early approaches relied on techniques such as synonym substitution (Topkara et al., 2006), syntactic structure restructuring (Atallah et al., 2001), or paraphrasing (Atallah et al., 2002). Later, advancements in modern neural language models led to improved methods that move away from rule-based approaches. Different approaches have been proposed, such as encoding messages by context-aware lexical substitution (Yang et al., 2022) or using mask-infilling models for editing text (Ueoka et al., 2021). Recent studies (Zhao et al., 2023; Kirchenbauer et al., 2023) explore modifying the logits of language models during token generation and embedding invisible watermarks in the decoding process. Our objective is to develop a robust watermarking technique for natural language models that maintain high text quality while effectively concealing identifying information.

**Post-hoc detection.** Rather than watermarking, an alternative approach involves developing detection models for post-hoc analysis of machine-generated text. Some detection methods use statistical outlier detection techniques without requiring additional training. For example, GLTR (Gehrmann et al., 2019) assesses the expected probability of individual tokens and applies thresholding to identify AI-generated content. DetectGPT (Mitchell et al., 2023) suggests that AI-generated passages tend to reside in the negative curvature of the log probability of texts. Another set of methods relies on classifiers that are fine-tuned to distinguish between human-written and machine-generated text. Initial efforts in this domain focus on detecting fake reviews (Hovy, 2016) and fake news (Zellers et al., 2019). More recently, OpenAI releases a web interface that uses a finetuned GPT model for this discrimination task (OpenAI, 2023b). However, as language models improve, AI-generated text is becoming increasingly similar to human-generated text, making it more challenging to detect. Gambini et al. (2022) find that existing detection strategies designed for GPT-2 struggle with GPT-3. Moreover, known detectors are found to be fragile to adversarial attacks (Wolff, 2020) and biased towards non-native English writers (Liang et al., 2023).

**Impossibility results?** Sadasivan et al. (2023) poses the question of whether detecting machine-generated text is possible and argue that as the human distribution and LLM distribution of texts get closer, any classifier will have to either have a large Type I error or a large Type II error. The authors also argue that (in Corollary 2) if the watermarking scheme can be learned then paraphrasing attacks either evade the detector or also classify humans with a similar distribution as false positives. This does not invalidate our results as we made no theoretical claim about paraphrasing. We do claim that in Theorem 3.1 that the watermarked LM $\hat{\mathcal{M}}$ and original LM $\mathcal{M}$ is statistically close — in fact, indistinguishable in the "differential privacy" sense. But the indistinguishability is for each token. As the number of tokens gets larger, they will eventually become distinguishable, that is why our Theorem C.4 and Theorem C.13 are not contradicting Theorem 3.1. This argument was initially pointed out by Chakraborty et al. (2023), showing that detection is possible.

**Language model watermarks with provable guarantees.** Concurrent to our work, Christ et al. (2023) consider the problem of formally defining watermarking language models and propose a construction with provable guarantees. The main differences between their work and ours are:

- In Christ et al. (2023), the watermarked distribution is computationally indistinguishable (i.e., indistinguishable against probabilistic polynomial-time algorithms) from the un-watermarked distribution whereas in our case, we insist that the watermarked distribution is statistically close to the un-watermarked distribution (of each token). The Type-I/Type-II error guarantees and the security properties are qualitatively different in both works.

- We both use different approaches to achieve our definitions. The advantage of our construction is that it satisfies robustness to edits property whereas they have no such guarantees. On the other hand, our construction uses a very different set of assumptions (e.g., high entropy) on the language model and prompt that appears to be incompatible with theirs.

- Finally, we implement our construction and conduct a thorough empirical evaluation to demonstrate its practicality while they don't provide any implementation of their construction.

**Statistical vs Cryptographic Watermarks.** Christ et al. (2023) and Aaronson (2023) are examples of *cryptographic* watermarks, while Kirchenbauer et al. (2023) and this paper study *statistical*

watermarks. There are several prominent differences that make it a bit challenging to compare the two kinds, but we will try. To start, we argue that both Christ et al. (2023) and Aaronson (2023) use a similar definition of language model watermarks as Definition 2.2 and considered a similar set of properties. Specifically, the "soundness", "completeness" from Christ et al. (2023) directly map to our "Type I error" and "Type II error" requirements. As we understand from the materials in Aaronson (2023)'s talk, their "indistinguishability" is a form of performance guarantee for $\hat{\mathcal{M}}$. The difference to ours is that they require (in our notation)

$$\mathbb{P}_{\hat{\mathcal{M}}(\text{prompt})}[\text{Next token}] = \mathbb{E}_{k}\left[\mathbb{P}_{\hat{\mathcal{M}}(\text{prompt})}[\text{Next token}|k]\right] = \mathbb{P}_{\mathcal{M}(\text{prompt})}[\text{Next token}]$$

where the random key k is marginalized out. while our results require that for every k the next token

$$\mathbb{P}_{\hat{\mathcal{M}}(\text{prompt})}[\text{Next token}|k] \approx_{\delta} \mathbb{P}_{\mathcal{M}(\text{prompt})}[\text{Next token}]$$

to be statistically close (in the same sense of $\delta$-differential privacy). By our metric, however, Aaronson (2023)'s watermark does not appear to satisfy any nontrivial $\delta$ guarantee, since it only requires *unbiasedness*. For that reason, the detection guarantee and its tradeoff with quality that we discussed in Remark C.15 is not applicable to the cryptographic watermarks.

## B    ADDITIONAL EXPERIMENT RESULTS

### B.1    EMPIRICAL ERROR RATES

We perform experiments on two datasets (OpenGen and LFQA) using three different models (GPT2-XL, OPT-1.3B, and LLaMA-7B). Table 4 presents the error rates, showcasing the sensitivity of the resulting hypothesis test based on observed $z$-scores. The results demonstrate that there are no Type-I (false positive) errors for all models, with true positive rates exceeding 0.94 for a threshold of $z = 6.0$.

| Dataset | Model | $z = 6.0$ | | | | $z = 7.0$ | | | |
|---|---|---|---|---|---|---|---|---|---|
| | | FPR | TNR | TPR | FNR | FPR | TNR | TPR | FNR |
| OpenGen | GPT2-XL | 0.0 | 1.0 | 0.943 | 0.057 | 0.0 | 1.0 | 0.832 | 0.168 |
| | OPT-1.3B | 0.0 | 1.0 | 0.998 | 0.002 | 0.0 | 1.0 | 0.996 | 0.004 |
| | LLaMA-7B | 0.0 | 1.0 | 0.974 | 0.026 | 0.0 | 1.0 | 0.911 | 0.089 |
| LFQA | GPT2-XL | 0.0 | 1.0 | 0.948 | 0.052 | 0.0 | 1.0 | 0.889 | 0.111 |
| | OPT-1.3B | 0.0 | 1.0 | 1.000 | 0.000 | 0.0 | 1.0 | 0.997 | 0.003 |
| | LLaMA-7B | 0.0 | 1.0 | 0.976 | 0.024 | 0.0 | 1.0 | 0.942 | 0.058 |

Table 4: Empirical error rates for watermark detection using different models on two datasets. All models employ nucleus sampling with $\delta = 2.0$ and $\gamma = 0.5$. No Type-I (false positive) errors are observed across all models.

### B.2    DIFFERENT WATERMARK PARAMETERS

We conduct an analysis to understand the impact of changing watermark strength ($\delta$), green list size ($\gamma$), and sampling methods on two datasets. The results are summarized in Table 5. When using nucleus sampling with a fixed $\gamma = 0.5$, increasing the watermark strength resulted in higher true positive rates (TPR), but it also led to an increase in perplexity (lower quality). Furthermore, for the same watermark strength $\delta$, varying the green list ratio from 0.25 to 0.5 and 0.75 showed improved detection results with smaller $\gamma$. Additionally, we explore different decoding methods, transitioning from nucleus sampling to multinomial sampling and beam search. Remarkably, watermark detection performed effectively with all decoding methods. It is worth noting that the perplexity score for beam search is significantly lower than that of nucleus sampling. However, beam search tends to generate shorter sequences with repeated words.

| Dataset | decoding | $\delta$ | $\gamma$ | PPL | $z = 6.0$ | | | | $z = 7.0$ | | | |
|---|---|---|---|---|---|---|---|---|---|---|---|---|
| | | | | | FPR | TNR | TPR | FNR | FPR | TNR | TPR | FNR |
| OpenGen | nucleus | 1.0 | 0.5 | $18.37_{6.45}$ | 0.0 | 1.0 | 0.576 | 0.424 | 0.0 | 1.0 | 0.310 | 0.690 |
| | nucleus | 2.0 | 0.5 | $19.42_{8.78}$ | 0.0 | 1.0 | 0.998 | 0.002 | 0.0 | 1.0 | 0.996 | 0.004 |
| | nucleus | 5.0 | 0.5 | $19.44_{15.02}$ | 0.0 | 1.0 | 1.000 | 0.000 | 0.0 | 1.0 | 1.000 | 0.000 |
| | nucleus | 10.0 | 0.5 | $19.20_{18.01}$ | 0.0 | 1.0 | 1.000 | 0.000 | 0.0 | 1.0 | 1.000 | 0.000 |
| | nucleus | 2.0 | 0.25 | $17.96_{9.54}$ | 0.0 | 1.0 | 1.000 | 0.000 | 0.0 | 1.0 | 1.000 | 0.000 |
| | nucleus | 2.0 | 0.75 | $20.03_{7.67}$ | 0.0 | 1.0 | 0.820 | 0.180 | 0.0 | 1.0 | 0.485 | 0.515 |
| | m-nom. | 2.0 | 0.5 | $1.75_{0.59}$ | 0.0 | 1.0 | 0.951 | 0.049 | 0.0 | 1.0 | 0.924 | 0.076 |
| | 4-beams | 2.0 | 0.5 | $1.83_{0.97}$ | 0.0 | 1.0 | 0.992 | 0.008 | 0.0 | 1.0 | 0.982 | 0.018 |
| | 6-beams | 2.0 | 0.5 | $1.89_{1.10}$ | 0.0 | 1.0 | 0.984 | 0.016 | 0.0 | 1.0 | 0.982 | 0.018 |
| | 8-beams | 2.0 | 0.5 | $1.96_{1.23}$ | 0.0 | 1.0 | 0.986 | 0.014 | 0.0 | 1.0 | 0.984 | 0.016 |
| LFQA | nucleus | 1.0 | 0.5 | $18.63_{7.19}$ | 0.0 | 1.0 | 0.455 | 0.545 | 0.0 | 1.0 | 0.199 | 0.801 |
| | nucleus | 2.0 | 0.5 | $19.14_{11.11}$ | 0.0 | 1.0 | 1.000 | 0.000 | 0.0 | 1.0 | 0.997 | 0.003 |
| | nucleus | 5.0 | 0.5 | $16.37_{15.39}$ | 0.0 | 1.0 | 1.000 | 0.000 | 0.0 | 1.0 | 1.000 | 0.000 |
| | nucleus | 10.0 | 0.5 | $16.07_{14.25}$ | 0.0 | 1.0 | 0.998 | 0.002 | 0.0 | 1.0 | 0.998 | 0.002 |
| | nucleus | 2.0 | 0.25 | $15.27_{10.00}$ | 0.0 | 1.0 | 1.000 | 0.000 | 0.0 | 1.0 | 1.000 | 0.000 |
| | nucleus | 2.0 | 0.75 | $19.44_{8.20}$ | 0.0 | 1.0 | 0.893 | 0.107 | 0.0 | 1.0 | 0.582 | 0.418 |
| | m-nom. | 2.0 | 0.5 | $3.17_{2.39}$ | 0.0 | 1.0 | 0.934 | 0.066 | 0.0 | 1.0 | 0.914 | 0.086 |
| | 4-beams | 2.0 | 0.5 | $3.24_{2.85}$ | 0.0 | 1.0 | 0.990 | 0.010 | 0.0 | 1.0 | 0.986 | 0.014 |
| | 6-beams | 2.0 | 0.5 | $3.20_{2.52}$ | 0.0 | 1.0 | 0.994 | 0.006 | 0.0 | 1.0 | 0.994 | 0.006 |
| | 8-beams | 2.0 | 0.5 | $3.13_{2.37}$ | 0.0 | 1.0 | 0.994 | 0.006 | 0.0 | 1.0 | 0.992 | 0.008 |

Table 5: Comparison of empirical error rates for watermark detection using nucleus sampling, multinomial decoding, and beam search. Each row represents the average of 500 sequences. While sequences generated with beam search exhibit lower perplexity, they tend to favor shorter outputs, potentially resulting in less diverse text.

## B.3 ADDITIONAL ROBUSTNESS RESULTS

In addition to the previously discussed robustness evaluations, we provide further analysis of our method's resilience against paraphrasing attacks and editing attacks. The results are presented in Figure 4. Notably, our proposed method (UNIGRAM-WATERMARK) consistently outperforms the baseline approach (KGW+23) across various datasets and attack scenarios. This demonstrates the superior robustness of our method in accurately detecting watermarked text.

## B.4 WHITE-BOX ATTACK

A potential attack for UNIGRAM-WATERMARK is to estimate the fixed green and red list. Then the adversary may attempt to bypass detection using these estimated lists. We conduct experiments on white-box attacks and we find that it is difficult to accurately estimate the green list. Even if the green list is known, our watermark is still somewhat effective thanks to our added robustness.

### B.4.1 ESTIMATING THE GREEN LIST TOKENS

The question arises: how can the adversary estimate the green list? We simulate an adversary attempting to learn the green list tokens by querying the model multiple times. The adversary collects token distributions from watermarked text and compares them to natural human distributions.

In our experiment, we query the LLaMA-13B watermarked model with watermark strength $\delta = 2.0$, watermark ratio $\gamma = 0.5$ (same setting in the paper) 2500 times, collecting 0.7 million tokens of watermarked text generated from the prompts in LFQA and OpenGen dataset.

Then we simulate three human data distributions:

1. The human response from the same prompt (LFQA and OpenGen dataset). The corresponding human output is 0.4 million tokens. We denote it as the "LFQA & OpenGen dataset"

2. Most times, human responses are not known. So we collect 2000 samples from the C4 (Raffel et al., 2020) dataset to form an approximate human dataset with 1 million tokens. We denote it as the "C4 dataset".

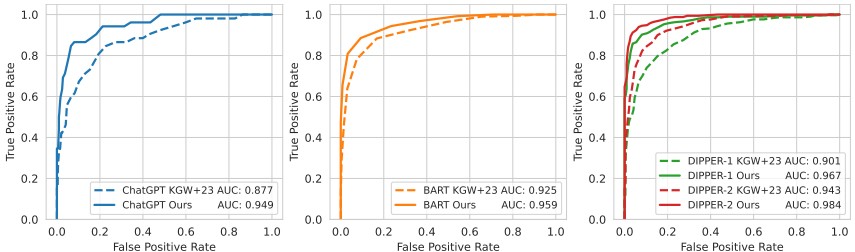

(a) UNIGRAM-WATERMARK against paraphrasing attacks on LFQA dataset with LLaMA-7B.

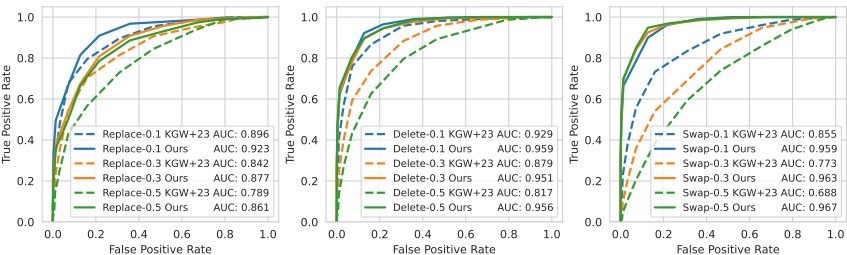

(b) UNIGRAM-WATERMARK against editing attacks on OpenGen dataset with LLaMA-7B. We vary the rates of synonym replacement, random deletion, and random swapping (0.1, 0.3, 0.5) to demonstrate different attack scenarios.

Figure 4: ROC curves with corresponding AUC values for watermark detection against various attack methods.

3. To simulate the distribution from non-native speakers. We also collect a non-native speaker (TOEFL essay) dataset from Liang et al. (2023) with 12k tokens. We denote it as the "Non-native dataset".

We calculate token frequencies for the three "human" datasets and the watermarked dataset. We use the following decision rule (Algorithm 3) to decide whether a token is green or red.

---

**Algorithm 3** Estimating the Green List tokens

---

1: **for** every token $v$ in the vocabulary $\mathcal{V}$ **do**
2: $\quad \Delta(v) \leftarrow$ Frequency($v$ in watermarked text) $-$ Frequency($v$ in human text)
3: $\quad$ **if** $\Delta(v) \geq 0$ **then**
4: $\quad\quad v$ is in the Green List.
5: $\quad$ **else**
6: $\quad\quad v$ is in the Red List.
7: $\quad$ **end if**
8: **end for**

---

The estimation results for the green list tokens are shown in the table below.

| Dataset | TPR | FPR | FNR | F1 |
|---|---|---|---|---|
| LFQA & OpenGen dataset | 0.692 | 0.830 | 0.170 | 0.755 |
| C4 dataset | 0.591 | 0.806 | 0.194 | 0.609 |
| Non-native dataset | 0.323 | 0.923 | 0.077 | 0.463 |

The results suggest that while it is possible to make non-trivial inferences about which token is green, it is hard to say for sure. Notice that we are using a rather big watermark strength. For smaller and more esoteric contexts (prompt, e.g., Non-native TOEFL dataset), such determination is harder.

### B.4.2 EVASION ATTACK (WHITE-BOX AND ESTIMATED)

In situations where the adversary has either an estimated version or full knowledge of the green and red lists, they can formulate an evasion strategy. We simulate this by assuming the adversary employs WordNet from NLTK to identify token synonyms. Tokens identified as in the green list are replaced with red list synonyms, noting that some tokens may not have synonyms or may only have green synonyms.

| Green List | Detect AUC | Avg PPL (eval by GPT-3) |
|---|---|---|
| No attack | 1.000 | 45.413 |
| Know all green tokens | 0.8413 | 193.410 |
| Estimated from LFQA & OpenGen dataset | 0.9397 | 189.423 |
| Estimated from C4 dataset | 0.9291 | 189.070 |
| Estimated from Non-native dataset | 0.9998 | 125.380 |

Table 6: Evasion attack results: analysis of detection AUC and perplexity.

The results in Table 6 show it is difficult to evade detection even with known green list tokens. The detection AUC for the watermarked text is still somewhat high. In addition, the honest attempt to evade the attack by automatic synonym replacement has led to a significant drop in the text quality.

### B.5 TESTING ON SCALED LANGUAGE MODELS

| | OpenGen | LFQA |
|---|---|---|
| **LLaMA-13B** | | |
| No attack | 1.000 | 1.000 |
| ChatGPT attack | 0.783 | 0.854 |
| **LLaMA-65B** | | |
| No attack | 1.000 | 1.000 |
| ChatGPT attack | 0.831 | 0.697 |

Table 7: Detection results (TPR at 1% FPR) for scaled models LLaMA-13B and LLaMA-65B.

We conduct supplementary experiments on the scaled models LLaMA-13B and LLaMA-65B. Using the same experimental settings as in the main paper, our preliminary results show that our method maintains effectiveness on these larger models. For LLaMA-13B, we are able to use the same test set size as in the original paper. For LLaMA-65B, due to computational constraints, we test on a sample of 100 sentences. The results (TPR at 1% FPR) are shown in Table 7.

### B.6 RESULTS FOR DEDUPLICATED DETECTION

| | OpenGen | LFQA |
|---|---|---|
| **LLaMA-13B** | | |
| No attack - Unique Detector | 1.000 | 1.000 |
| ChatGPT attack - Unique Detector | 0.679 | 0.773 |
| **LLaMA-65B** | | |
| No attack - Unique Detector | 1.000 | 1.000 |
| ChatGPT attack - Unique Detector | 0.783 | 0.682 |

Table 8: Detection results (TPR at 1% FPR) with "Unique" detector.

An alternative detector, named "Unique" demonstrates improved robustness in detection and offers advantages in controlling false positives with ease (Section E). We conduct experiments to evaluate deduplicated detection performance, with the outcomes presented in Table 8.

## C MAIN THEORETICAL RESULTS WITH PROOFS

In this section, we state and prove the guarantees for UNIGRAM-WATERMARK which certifies the required quality, correctness, and security properties of a language model watermarking scheme from Definition 2.2.

**Symbols and mathematical notations.** We use $\mathbb{P}[\cdot]$, $\mathbb{E}[\cdot]$, $\mathbb{P}[\cdot|\cdot]$ and $\mathbb{E}[\cdot|\cdot]$ to denote the probability, expectation operator, conditional probability and conditional expectation respectively. Whenever there is ambiguity on which distribution the random variables are drawn from, we explicitly state them, e.g., $\mathbb{P}_{(X,Y)\sim\mathcal{D}}[X < 3|Y = y]$, or equivalently $\mathbb{P}[X < 3|Y = y \; ; \; (X,Y) \sim \mathcal{D}]$. To avoid clutter, we do not distinguish between random variables and constants as the distinctions are clear from the context. Boldface symbols denote a vector, e.g., a probability mass function $\mathbf{p}$ or a sequence of tokens $\boldsymbol{y}$. $\| \cdot \|_2, \| \cdot \|_\infty$ denotes the standard $\ell_2$ and $\ell_\infty$-norms of a vector. In addition, $[n]$ is a shorthand for $\{1, 2, ..., n\}$. Other symbols and their meanings will be defined as we encounter them.

### C.1 QUALITY GUARANTEES

We start by providing a strong utility analysis of the watermarked language model than the "perplexity" bound from (Kirchenbauer et al., 2023). Our results work for the entire family of Rényi-divergence and imply guarantees in Kullback-Leibler (KL) divergence and Total Variation-distance.

The Renyi-divergence of two distributions $P$, $Q$ is defined as

$$D_\alpha\big(P\|Q\big) = \frac{1}{\alpha - 1} \log \mathbb{E}_{x\sim Q} \left[ \big(\frac{dP}{dQ}\big)^\alpha \right]$$

where $\frac{dP}{dQ}$ is the Radon–Nikodym derivative. When $\alpha \to 1$, the Renyi divergence converges to the KL-divergence. Additionally, when $\alpha = 0.5$, it serves as an upper bound for the TV-distance.

On the technical level, we leverage a surprising connection to a modern machinery developed in the differential privacy literature known as "bounded range" analysis (Dong et al., 2020) of the classical exponential mechanism (McSherry and Talwar, 2007).

**Theorem C.1** (Restatement of Theorem 3.1). *Consider $\boldsymbol{h}$ as the input to the language model at step $t$, denoted as $\boldsymbol{h} = [\boldsymbol{x}, \boldsymbol{y}_{1:t-1}]$. Fix green list $G$. Let $\delta$ represent the watermark strength. For any $\boldsymbol{h}$, the $\alpha$-th order Renyi-divergence between the watermarked probability distribution $\hat{\mathbf{p}}_t = \hat{\mathbf{p}}_t(\cdot|\boldsymbol{h})$ at time step $t$ and the original probability distribution $\mathbf{p}_t = \mathbf{p}_t(\cdot|\boldsymbol{h})$ satisfies:*

$$\forall \boldsymbol{h}, \max\big(D_\alpha\big(\hat{\mathbf{p}}_t\|\mathbf{p}_t\big), D_\alpha\big(\mathbf{p}_t\|\hat{\mathbf{p}}_t\big)\big) \leq \min\{\delta, \alpha\delta^2/8\}.$$

*Proof.* We define $\delta_v = 0$ when $v \in R$ and $\delta_v = \delta$ when $v \in G$. Using this definition, we have:

$$\hat{\mathbf{p}}(v|\boldsymbol{h}) = \frac{\exp(\boldsymbol{\ell}_v + \delta_v)}{\sum_w \exp(\boldsymbol{\ell}_w + \delta_w)} \leq \frac{\exp(\delta)\exp(\boldsymbol{\ell}_v)}{\exp(-\delta)\sum_w \exp(\boldsymbol{\ell}_w)} = e^{2\delta}\mathbf{p}(v|\boldsymbol{h})$$

Similarly, $\hat{\mathbf{p}}(v|\boldsymbol{h}) \geq e^{-2\delta}\mathbf{p}(v|\boldsymbol{h})$.

Consequently, $\hat{\mathbf{p}}$ and $\mathbf{p}$ are $2\delta$-close in terms of max-divergence, which can be interpreted as $(\epsilon, \tilde{\delta})$-indistinguishable, similar to the concept of Differential Privacy (Dwork et al., 2006) with $\tilde{\delta} = 0$ and $\epsilon = 2\delta$.

Additionally, $\hat{\mathbf{p}}(v|\boldsymbol{h})$ and $\mathbf{p}(v|\boldsymbol{h})$ satisfy $\delta$-BoundedRange (Proposition 1 in Dong et al. (2020)) with parameter $\delta$, since the changes to $\boldsymbol{\ell}_v$ is monotonic. Lemma 3.2 in Cesar and Rogers (2021) shows that $\delta$-Bounded Range implies $\delta^2/8$-concentrated differential privacy, which says that $D_\alpha(\hat{\mathbf{p}}\|\mathbf{p}) \leq \frac{\delta^2\alpha}{8}$ for all $\alpha \geq 1$ (where $D_\alpha$ represents Rényi Divergence of order $\alpha$). Specifically, when $\alpha = 1$, the KL-divergence satisfies $D_{\mathrm{KL}}(\hat{\mathbf{p}}\|\mathbf{p}) \leq \frac{\delta^2}{8}$.

Furthermore, $\delta$-BoundedRange implies $\delta$-DP (or rather $(\delta, 0)$-indistinguishability, since we are dealing with just two distributions rather than a family of neighbor distributions). It follows from the that

$$D_{\mathrm{KL}}(\hat{\mathbf{p}}\|\mathbf{p}) \leq D_\infty(\hat{\mathbf{p}}\|\mathbf{p}) \leq \delta$$

$\square$

**Corollary C.2.** *For any prompt $\boldsymbol{x}$, the KL-divergence between the probability distribution of the watermarked sequence and the original sequence satisfies:*

$$\forall \boldsymbol{x}, \max\{D_{\mathrm{KL}}\big(\hat{\mathbf{p}}(\boldsymbol{y}_{1:n}|\boldsymbol{x})\|\mathbf{p}(\boldsymbol{y}_{1:n}|\boldsymbol{x})\big), D_{\mathrm{KL}}\big(\mathbf{p}(\boldsymbol{y}_{1:n}|\boldsymbol{x})\|\hat{\mathbf{p}}(\boldsymbol{y}_{1:n}|\boldsymbol{x})\big)\} \leq \alpha \min\{n\delta, n\delta^2/8\}$$

*Proof.* The proof follows from the adaptive composition theorem for Renyi-divergence, and max-divergence (from the DP literature) for the autoregressive decomposition of $\hat{\mathbf{p}}(\boldsymbol{y}_{1:n}|\boldsymbol{x})$ and $\mathbf{p}(\boldsymbol{y}_{1:n}|\boldsymbol{x})$ and then invoke Theorem 3.1 for each factor. □

## C.2 ROBUSTNESS / SECURITY GUARANTEES

In this section, we provide the proof for Theorems 3.7, D.1, and 3.1 to ensure completeness and precision. We begin by restating the theorems and providing the corresponding proofs with necessary modifications.

**Theorem C.3** (Robustness to editing (Restatement of Theorem 3.7) ). *Let $\boldsymbol{y} = [y_1, \ldots, y_n]$ represent the watermarked sequence. Suppose the adversary $\mathcal{A}$ follows Definition 2.2 and outputs a modified text $\boldsymbol{u} = [u_1, \ldots, u_m]$. Following Equation 2, we calculate z-score $z_{\boldsymbol{y}}$ and $z_{\boldsymbol{u}}$. Assume edit distance between $\boldsymbol{y}$ and $\boldsymbol{u}$ (denoted as $\eta$) satisfies $\eta < n$. Then we have*

$$z_{\boldsymbol{u}} \geq z_{\boldsymbol{y}} - \max\{\frac{(1+\gamma/2)\eta}{\sqrt{n}}, \frac{(1-\gamma/2)\eta}{\sqrt{n-\eta}}\}.$$

*In particular, when $\eta \leq \frac{2\gamma n}{(1+\gamma/2)^2}$, we can drop the second term in the max.*

*Proof.* Define bivariate function $f(x,y) = \frac{x-\gamma y}{\sqrt{y}}$. By Taylor's theorem

$$f(x-k_x, y-k_y) = f(x,y) + \begin{bmatrix} \partial_x f(x - \tilde{k}_x y - \tilde{k}_y) \\ \partial_y f(x - \tilde{k}_x y - \tilde{k}_y) \end{bmatrix}^T \begin{bmatrix} -k_x \\ -k_y \end{bmatrix} = f(x,y) - \left( \frac{k_x}{\sqrt{y-\tilde{k}_y}} - \frac{\gamma k_y}{2\sqrt{y-\tilde{k}_y}} \right)$$

where $\tilde{k}_x$ is between $0$ and $k_x$ and $\tilde{k}_y$ is between $0$ and $k_y$. We also know that $|k_x| \leq k$ and $|k_y| \leq k$.

A lower bound of the above can be obtained by finding an upper bound to

$$\frac{k_x}{\sqrt{y-\tilde{k}_y}} - \frac{\gamma k_y}{2\sqrt{y-\tilde{k}_y}} = \frac{k_x - \frac{\gamma}{2}k_y}{\sqrt{y-\tilde{k}_y}}$$

First observe that we can always choose $k_x = k$. Next we discuss two possibilities of $k_y$. If $k_y$ is negative, then choosing $k_y = -k$ and $\tilde{k} = 0$ maximizes the bound, which gives $\frac{(1+\gamma/2)k}{\sqrt{y}}$.

If $k_y$ is positive, then we should always choose $\tilde{k}_y = k_y$ to maximize the expression, which gives us an upper bound of

$$\frac{k - \frac{\gamma}{2}k_y}{\sqrt{y-k_y}} = \frac{k + \frac{\gamma}{2}(y-k_y) - \frac{\gamma}{2}y}{\sqrt{y-k_y}} = \frac{k - \frac{\gamma}{2}y}{\sqrt{y-k_y}} + \frac{\gamma\sqrt{y-k_y}}{2}.$$

We will discuss two cases again, the first case is when $k - \gamma y/2 \leq 0$. In this case, the function $g(u) = a/u + bu$ with $a \leq 0$ has a derivative of $-a/u^2 + b \geq 0$, thus $g$ is monotonically increasing. Thus we should choose $k_y = 0$. The second case is when $k - \gamma y/2 > 0$, in this case the $a > 0$ in the above $g(u)$ and $g(u)$ is convex, thus $\max_{u_{\min} \leq u \leq u_{\max}} g(u) = \max\{g(u_{\max}), g(u_{\min})\}$. Thus we should just compare the two cases when $k_y = 0$ and $k_y = k$, i.e., $\max\{\frac{k}{\sqrt{y}}, \frac{(1-\gamma/2)k}{\sqrt{y-k}}\}$.

Collect everything together, we get an upper bound o

$$\max\{\frac{(1+\gamma/2)k}{\sqrt{y}}, \frac{k}{\sqrt{y}}, \frac{(1-\gamma/2)k}{\sqrt{y-k}}\} = \max\left\{ \frac{(1+\gamma/2)k}{\sqrt{y}}, \frac{(1-\gamma/2)k}{\sqrt{y-k}} \right\}$$

i.e.,

$$f(x-k_x, y-k_y) - f(x,y) \geq -\max\left\{ \frac{(1+\gamma/2)k}{\sqrt{y}}, \frac{(1-\gamma/2)k}{\sqrt{y-k}} \right\}.$$

Now notice that our $z$-score has the same form as the $f(x, y)$ function. We can take $y = n$ and $x = |\boldsymbol{y}|_G$. Instantiate $k$ be the maximum number of edits $\eta$. Observe that given that the adversary has a bounded edit distance, each operation of "insertion", "deletion", or "edit" can, at most, alter one token from the green list to the red list. They also can only alter the length by the number of edits. The above result translates into

$$z_{\boldsymbol{u}} \geq z_{\boldsymbol{y}} - \max\{\frac{(1 + \gamma/2)\eta}{\sqrt{n}}, \frac{(1 - \gamma/2)\eta}{\sqrt{n - \eta}}\},$$

where $\eta$ denotes the edit distance between $y$ and $u$. □

The robustness theorem above implies the security guarantees as we discussed in Corollary C.23.

### C.3 No false positive (Type I error guarantees)

**Theorem C.4** (No false positives ). *Consider $\boldsymbol{y} = \boldsymbol{y}_{1:n}$ as any* fixed *suspect text. Let $N =: |\mathcal{V}|$ and $G \subset |\mathcal{V}|$ satisfying $|G| = \gamma N$. $G$ is selected through Algorithm 1, using a uniform random choice. Let $|\boldsymbol{y}|_G$ denote the number of tokens in $G$ and $z_{\boldsymbol{y}} := \frac{|\boldsymbol{y}|_G - \gamma n}{\sqrt{n\gamma(1-\gamma)}}$ as in Algorithm 2. Then the following statements hold true:*

*1. Assume $n \geq 1$, then*
$$\mathbb{E}[|\boldsymbol{y}|_G | \boldsymbol{y}] = \gamma n \quad and \quad \mathbb{E}[z_{\boldsymbol{y}} | \boldsymbol{y}] = 0.$$

*2. Define $C_{\max}(\boldsymbol{y}) := \max_{i \in [N]} \sum_{j=1}^{n} \mathbf{1}(y_j = i)$ and $V(\boldsymbol{y}) := \frac{1}{n} \sum_{i=1}^{N} (\sum_{j=1}^{n} \mathbf{1}(y_j = i))^2$, then with probability $1 - \alpha$ (over only the randomness of $G$),*

$$\mathbb{P}\left[|\boldsymbol{y}|_G \geq \gamma n + \sqrt{64\gamma nV \log(9/\alpha)} + 16C_{\max}\log(9/\alpha) \Big| \boldsymbol{y}\right] \leq \alpha$$

*or equivalently (when $n \geq 1$)*

$$\mathbb{P}\left[z_{\boldsymbol{y}} \geq \sqrt{\frac{64V \log(9/\alpha)}{1 - \gamma}} + \frac{16C_{\max}\log(9/\alpha)}{\sqrt{n\gamma(1-\gamma)}} \Big| \boldsymbol{y}\right] \leq \alpha.$$

*Proof.* To prove the first statement, observe that any fixed token has a probability $\gamma$ to be included in the green list, thus by the linearity of the expectation and the independence of $\boldsymbol{y}$ in $G$.

$$\mathbb{E}[|\boldsymbol{y}|_G | \boldsymbol{y}] = \sum_{i=1}^{n} \mathbb{E}[\mathbf{1}(y_i \in G) | \boldsymbol{y}] = \sum_{i=1}^{n} \gamma = \gamma n.$$

Next, we will prove the second statement by applying Lemma F.1 to obtain the result stated in the third statement. Let $a_{i,j} = \mathbf{1}(j \leq \gamma N) \sum_{\ell=1}^{n} \mathbf{1}(y_\ell = i)$. By our assumption $0 \leq a_{i,j} \leq C_{\max}$ for all $i, j$. Observe that $\sum_{i=1}^{N} a_{i,\Pi_N(i)}$ is *identically distributed* with $|\boldsymbol{y}|_G$.

By Lemma F.1 with $t = 16\log(8e^{1/16}/\alpha)$, we get that with probability $1 - \alpha$,

$$||\boldsymbol{y}|_G - \gamma n| < 2\sqrt{\frac{16\log(9/\alpha)}{N} N\gamma nV} + 16C_{\max}\log(9/\alpha)$$

where we used that $8e^{1/16} \leq 9$ and the fact that only $\gamma N$ columns of the $a_{i,j}$ matrix $a_{i,j}$ is nonzero, and for each non-zero column L2-norm of the column is bounded by $\sqrt{nV}$ by our definition of $V$. The result for the $z$-score follows trivially. □

*Remark* C.5 (Wide applicability). Note that the theorem does not impose assumptions on how $\boldsymbol{y}$ is generated. It covers any procedure (including human generation) that produces $\boldsymbol{y}$ in a manner *independently* of the secret partition $G$. In cases where $\boldsymbol{y}$ is generated by a language model, it could be the output of greedy search from $\mathbf{p}(y_t | \boldsymbol{x}, \boldsymbol{y}_{1:t-1})$, nucleus sampling, beam search, or any other decoding methods.

*Remark* C.6 (Diversity parameters). The $V$ and $C_{\max}$ parameters in Theorem C.4 measure the *diversity* of the suspect text $\boldsymbol{y}$ and are necessary for the high-probability bound. As an example, if the prompt says "`Repeat "Goal" for a hundred thousand times like a soccer commentator.`" Then the resulting generated sequence will be "`Goal goal goal ...`", and has either $n$ green tokens or $0$ green tokens. No meaningful Type I error bound can be obtained.

*Remark* C.7 (Controlling false positive rate). The theorem implies that if we choose $\tau > \sqrt{\frac{64V \log(9/\alpha)}{1-\gamma}} + \frac{16C_{\max}\log(9/\alpha)}{\sqrt{n\gamma(1-\gamma)}}$, then the false-positive rate is smaller than $\alpha$. Note that $V$ and $C_{\max}$ can be computed directly from $\boldsymbol{y}$, allowing us to choose an input-dependent $\tau$ as a function of $V, C_{\max}$ that achieves a $\alpha$-Type I error guarantee with a fixed $\alpha$ for all inputs. In particular, the Type I error $\alpha$ decreases exponentially as we increase the threshold $\tau$.

## C.4 ONLY TRUE DETECTION (TYPE II ERROR GUARANTEES)

For bounding the Type II error, i.e., false negative rates, we will work with our proposed method that generates $\boldsymbol{y}$ from the language model, i.e., sampling from the watermarked distribution $\hat{\mathbf{p}}$ recursively one token at a time.

Let's first recall a few notations. $\boldsymbol{h}$ is the input to the language model at step $t$, i.e., $\boldsymbol{h} = [\boldsymbol{x}, \boldsymbol{y}_{1:t-1}]$. Let $\delta$ represent the watermark strength from Equation 1. The green list $G \subset [N]$ is a random index set of the vocabulary of size $\gamma N$. The watermarked probability distribution $\hat{\mathbf{p}}_t = \hat{\mathbf{p}}_t(\cdot|\boldsymbol{h})$ at time step $t$. The process of generating the sentence $y_1, y_2, \ldots, y_n$ involves recursively sampling from $\hat{\mathbf{p}}_t$, which we refer to as a "roll-out" procedure.

We need to make a few assumptions about the language model's probability distribution $\mathbf{p}$ and the prompt $\boldsymbol{x}$. We will first state them and then explain why these are natural and arguably needed for the Type II error to be small.

### C.4.1 ON-AVERAGE HIGH ENTROPY ASSUMPTION

The first such assumption requires the probability of the roll-out to be "sufficiently diverse" on average. We will introduce the notation $\|\mathbf{p}\|_2 := \sqrt{\sum_{i=1}^{N} \mathbf{p}[i]^2}$.

**Assumption C.8** (On-average-high-entropy). We say a language model's probability distribution $\mathbf{p}$ with a prompt $\boldsymbol{x}$ satisfies $\xi$-on-average-high-entropy if

$$\frac{1}{n}\sum_{t=1}^{n} \mathop{\mathbb{E}}_{\boldsymbol{y}_{1:t-1}\sim\mathbf{p}(\cdot|\boldsymbol{x})}[\|\mathbf{p}_t\|^2] \le \xi.$$

This assumption requires the distribution of the roll-out to be sufficiently diffuse on average (either in expectation or with high probability).

The purpose of these assumptions is to rule out the cases when $\boldsymbol{y}_{1:n}$ is almost deterministic under $\mathbf{p}$ and perturbing the logits by $\delta$ does not change the distribution much at all.

For example, if the prompt writes

"`Generate the English alphabet in capital letters for 200 times please.`"

Then the language model would generate

"`ABC...XYZ, ABC...XYZ, ...`".

Despite that the generated sequence is very long, i.e., $n$ is as large as $5,200$, the added watermark does not change the distribution very much at all. To see this, if $\mathbf{p}(y_3 = \text{"C"}|\boldsymbol{x}, \boldsymbol{h}) \ge 1 - \epsilon$ for a tiny $\epsilon$, and then by our quality guarantee, $\hat{\mathbf{p}}(y_3 = \text{"C"}|\boldsymbol{x}, \boldsymbol{h}) \ge 1 - \epsilon e^{\delta}$.

Quantitatively, for nearly uniform $\mathbf{p}_t$, $\xi = O(1/N)$, if $\mathbf{p}_t$ concentrates on a single token for all $t$, e.g., when a football commentator exclaims "`Goal goal goal goal ....`", then we cannot obtain a better bound than the trivial $\xi \le 1$. In the alphabet example above $\xi \le 1/26$.

**Why is it called entropy?** Assumption C.8 is related to the "high-entropy" assumption in Kirchenbauer et al. (2023) but for a slightly different kind of entropy. In a more formal sense, the quantity $\|\mathbf{p}_t\|^2$ is connected to the Tsallis entropy of order 2, defined as $S_2(\mathbf{p}_t) = k_B(1 - \|\mathbf{p}_t\|^2)$ where $k_B$ is known as the Boltzmann constant. Our assumption requires the expected Tsallis entropy of the conditional distribution $\mathbf{p}_t$ over the roll-out of $\mathbf{p}$ to be larger than $k_B(1 - \xi)$ on average among $t = 1, ..., n$.

For a high-probability result, we also need a stronger version.

**Assumption C.9** (On-average-high-entropy (high probability)). We say that a language model's probability distribution $\mathbf{p}$ with a prompt $\boldsymbol{x}$ satisfies $(\xi, \beta)$-on-average-high-entropy if with probability at least $1 - \beta$ over the generated sequence $\boldsymbol{y}_{1:n}$,

$$\frac{1}{n} \max \left\{ \left\| \sum_{t=1}^n \mathbf{p}_t \right\|, \sum_{t=1}^n \|\mathbf{p}_t\|^2, \left\| \sum_{t=1}^n \mathbf{p}_t \right\|_\infty, \sum_{t=1}^n \|\mathbf{p}_t\|_\infty^2 \right\} \leq \xi.$$

The behavior is similar to that of the expectation version of the assumption. When $\mathbf{p}_t$ is nearly uniform, $\mathbf{p}_t[i] = O(1/N)$, then $\xi = O(1/\sqrt{N})$. When $\mathbf{p}_t$ is supported only on one token, then $\xi = 1$. In practice, $\xi$ is a small constant. As we will present in the main theorem, as long as $\xi \asymp \delta$, the number of green list tokens is guaranteed to grow faster $\gamma n$ as $n$ gets larger.

One may also ask whether it is necessary to make entropy assumptions on the conditional probabilities instead of the marginal probabilities induced by $\mathbf{p}$ or $\hat{\mathbf{p}}$, but this is unfortunately not sufficient as illustrated in the following example.

**Example C.10** (Marginal high entropy is insufficient). Let the prompt $\boldsymbol{x}$ be

```
"Generate the first token uniformly at random, then repeat the
        token you generated for the remaining n − 1 tokens".
```

In this case, a good language model that follows the instruction will have $\mathbb{P}_{\mathbf{p}}(y_t = i) = 1/N$ for all $i$ and all $t = 1, ..., n$ marginally, which implies that the entropy is the maximum and for any green list $G$, $\mathbb{P}_{\mathbf{p}}(y_t \in G) = \gamma$. On the other hand, with probability $\gamma$, $|\boldsymbol{y}|_G = n$ and with probability $1 - \gamma$, $|\boldsymbol{y}|_G = 0$. There isn't any concentration around $\gamma n$ possible. Moreover, check that if we apply watermark, then $\mathbb{P}_{\hat{\mathbf{p}}}(y_t \in G) = \frac{\gamma e^\delta}{\gamma e^\delta + (1-\gamma)}$ for all $t$ and all $G$. This changes the probability of seeing $|\boldsymbol{y}|_G = n$ slightly but the two world remains indistinguishable.

### C.4.2 A "HOMOPHILY" ASSUMPTION

The second assumption that we need to make is called "homophily", which says that increasing the probability of a group of tokens by adding the watermarks will not decrease the probability of generating the same group of tokens in the future as the language model rolls out.

**Assumption C.11** ("Homophily"). We say a language model's probability distribution $\mathbf{p}$ and prompt $\boldsymbol{x}$ satisfy "homophily" if for any $G$, the corresponding watermarked $\hat{\mathbf{p}}$ satisfies that

$$\mathbb{E}_{\boldsymbol{h} \sim \hat{\mathbf{p}}(\cdot|\boldsymbol{x})} \left[ \mathbb{P}_{y \sim \hat{\mathbf{p}}(\cdot|\boldsymbol{h},\boldsymbol{x})} (y \in G) \right] \geq \mathbb{E}_{\boldsymbol{h} \sim \mathbf{p}(\cdot|\boldsymbol{x})} \left[ \mathbb{P}_{y \sim \hat{\mathbf{p}}(\cdot|\boldsymbol{h},\boldsymbol{x})} (y \in G) \right]$$

where $\boldsymbol{h}$ denotes the generated sequence before $y$.

This assumption says that by increasing the probability of tokens in $G$, the induced distribution of the prefix $\boldsymbol{h}$ cannot counter-intuitively reduce the probability of tokens in $G$ in the future on average.

The assumption is not unreasonable, because we expect a language model to be more likely to refer to text it has generated in the prefix than those that did not appear in the prefix.

This "homophily" assumption is needed to rule out the unnatural situation where increasing the green list tokens initially ends up reducing the number of green list tokens in the long run. To illustrate this, consider the following example utilizing the prompt:

$\boldsymbol{x} =$ "Randomly select a color, state what it is. Then write a short
        poem about it without naming this color at all."

The generated text from a commercial language model is

```
    "Color choice:  green.   Emerald whispers in the meadow's sway,
  Life's verdant rhythm in ceaseless play.   It cradles the world in
             a leafy embrace, A silent serenade to nature's grace."
```

Notice that if the token "green" $\in G$, it increases the probability of the language model generating "green" at the beginning. However, regardless of the text's length, the subsequent portion of the generated text will not contain the word "green", as instructed by the prompt. This decreases the expected number of times the token "green" appears.

To hammer it home, consider the following more quantitative construction of that works no matter which random green list $G$ realizes.

$$\boldsymbol{x} = \text{``Choose the first k token by random sampling without replacement.   Then sample from all but the token you choose uniformly for n-k rounds.''}$$

It's easy to calculate that the expected number of times any token appears in a language model that perfectly follows the instruction will be $n/N$. However, the watermarked language model, let's say we use a very large $\delta$ such that the first $k$ tokens are from the green list, then the expected number of times a green-list token appears is $\frac{k}{\gamma N} + \frac{\gamma N - k}{\gamma N} \frac{(n-k)(\gamma N - k)}{N-k}$ which is bounded by 1 if $k = \gamma N$ instead of growing linearly in $n$ as in the original language model.

To obtain a concentration bound, we also need a stronger version of the homophily assumption as follows.

**Assumption C.12** (High probability on-average homophily). There exists a coupling – a joint distribution of $\boldsymbol{y}_{1:n}$ and $\hat{\boldsymbol{y}}_{1:n}$ where marginally $\boldsymbol{y}_{1:n} \sim \mathbf{p}(\cdot|\boldsymbol{x})$, $\hat{\boldsymbol{y}}_{1:n} \sim \hat{\mathbf{p}}(\cdot|\boldsymbol{x})$ – such that for any $G$, with probability $1 - \beta$ over the joint distribution,

$$\frac{1}{n} \sum_{t=1}^n \hat{\mathbf{p}}_t(G|\hat{\boldsymbol{y}}_{1:t-1})) \geq \frac{1}{n} \sum_{t=1}^n \hat{\mathbf{p}}_t(G|\boldsymbol{y}_{1:t-1})).$$

The reason for defining the existence of a coupling is for technical reasons, but the purpose of the assumption is identical to that of the in-expectation version.

### C.5 THEOREM STATEMENT ON "ONLY TRUE DETECTION"

Now we are ready to state the main theorem.

**Theorem C.13** (Only true detection). *For a fixed language model $\mathcal{M}$ and a prompt $\boldsymbol{x}$. The sentence $\boldsymbol{y}_{1:n}$ generated from $\hat{\mathcal{M}}(\boldsymbol{x})$ where $\hat{\mathcal{M}}$ is an output of our watermarking scheme $\mathsf{Watermark}_{\delta,\gamma}(\mathcal{M})$ with parameter $\delta, \gamma$. Then the following statements are true.*

*1. Assume homophily (Assumption C.11), then*

$$\mathbb{E}[|\boldsymbol{y}|_G] \geq \frac{n\gamma e^\delta}{1 + (e^\delta - 1)\gamma} - \gamma(1-\gamma)e^\delta \sum_{t=1}^n \mathop{\mathbb{E}}_{\boldsymbol{y}_{1:t-1} \sim \mathbf{p}(\cdot|\boldsymbol{x})} \|\mathbf{p}_t\|^2.$$

*In particular, if Assumption C.8 condition is true with parameter $\xi \leq (1 - \kappa)\frac{e^\delta - 1}{(1+(e^\delta-1)\gamma)e^\delta}$ for a parameter $0 < \kappa < 1$, then*

$$\mathbb{E}[|\boldsymbol{y}|_G] \geq n\gamma \left(1 + \kappa \frac{(e^\delta - 1)(1 - \gamma)}{1 + (e^\delta - 1)\gamma}\right) \text{ or equivalently } \mathbb{E}[z_{\boldsymbol{y}}] \geq \frac{\kappa(e^\delta - 1)\sqrt{n\gamma(1 - \gamma)}}{1 + (e^\delta - 1)\gamma}.$$

*2. Assume high-probability version of homophily (Assumption C.12). There exists a parameter $C_{\delta,\gamma}$ that depends only $\delta, \gamma$ such that with probability at least $1 - \beta$ for any $\beta > 0$ (over both $G$ and*

$y \sim \hat{\mathbf{p}}(\cdot | \boldsymbol{x}, G)$ ),

$$\|\boldsymbol{y}\|_G \geq \frac{n\gamma e^{\delta}}{1 + (e^{\delta} - 1)\gamma} - \sqrt{2n \log(6/\beta)}$$
$$- C_{\delta,\gamma} \log^2 \frac{27(n+1)}{\beta} \left( \|\sum_{t=1}^n \mathbf{p}_t\| + \sum_{t=1}^n \|\mathbf{p}_t\|^2 + \|\sum_{t=1}^n \mathbf{p}_t\|_{\infty} + \sum_{t=1}^n \|\mathbf{p}_t\|_{\infty}^2 \right).$$

*In particular, if for a parameter $0 < \kappa < 1$,*

$$n \geq \frac{8 \log(6/\beta)(1 - \gamma + e^{\delta}\gamma)^2}{(1 - \kappa)^2 \gamma^2 (1 - \gamma)^2 (e^{\delta} - 1)^2} = \tilde{\Omega}(1/\delta^2) \tag{3}$$

*and Assumption C.9 condition is true with parameter $(\xi, \beta/3)$ where*

$$\xi \leq \frac{(1 - \kappa)\gamma(1 - \gamma)(e^{\delta} - 1)}{8 C_{\delta,\gamma}(1 - \gamma + e^{\delta}\gamma) \log^2 \left( \frac{27(n+1)}{\beta} \right)} = \tilde{O}(\delta), \tag{4}$$

*then*

$$\mathbb{P}\left[\|\boldsymbol{y}\|_G < n\gamma(1 + \kappa \frac{(e^{\delta} - 1)(1 - \gamma)}{1 - \gamma + \gamma e^{\delta}})\right] = \mathbb{P}\left[z_{\boldsymbol{y}} < \frac{\kappa(e^{\delta} - 1)\sqrt{n\gamma(1 - \gamma)}}{1 + (e^{\delta} - 1)\gamma}\right] \leq \beta.$$

*Remark* C.14 (Exponentially small Type I and Type II error guarantees). Recall that according to Theorem C.4, in order to have a false positive rate controlled at level $\alpha$, we need to set the threshold $\tau \gtrsim \sqrt{\log(1/\alpha)}$ for sufficiently high-entropy sequences. Theorem C.13 says that if we want the false negative rate to be smaller than $\beta$, we only need the threshold $\tau \lesssim \kappa\delta n$ under similar (slightly different) high-entropy sequences for $n \gtrsim \log(1/\beta)/\delta^2$. Observe that there is a wide range of valid choices of $\tau$ for us to have a detection algorithm that does not make Type I or Type II error with high probability. These observations together suggest that we can afford to choose $\delta \asymp 1/\sqrt{n}$ if the sequence is sufficiently high-entropy.

*Remark* C.15 (Information-theoretic optimality). The sample complexity of $n \gtrsim 1/\delta^2$ is information-theoretically optimal (up to a logarithmic factor) in $\delta$ because, our accuracy guarantee (together with the composition theorem) indicates that the KL-divergence between a sequence of length $n$ generated from $\mathbf{p}$ and that generated from $\hat{\mathbf{p}}$ is $n\delta^2$ indistinguishable, i.e., $n > 1/\delta^2$ for any classifier — even the uniform most-powerful Neyman-Pearson likelihood-ratio test (which requires additional information, e.g., $\boldsymbol{x}$ and $\mathbf{p}$ which we do not have) — to make no mistakes with a constant probability.

## C.6 PROOF OF THEOREM C.13

In the false negative error cases, $\boldsymbol{y}$ is drawn from the watermarked language model $\hat{\mathcal{M}}$. To be explicit, let us write $\boldsymbol{y} = [\hat{y}_1, ..., \hat{y}_n] = \hat{\boldsymbol{y}}_{1:n}$. Now let's also define a hypothetical (possibly coupled) sequence $\boldsymbol{y}_{1:n}$ which is drawn from the original (un-watermarked) language model $\mathcal{M}$.

For convenience, we define the following shorthand $\mathbf{p}(G) := \mathbb{P}_{y \sim \mathbf{p}}[y \in G]$. for a probability mass function $\mathbf{p}$ defined on the vocabulary $\mathcal{V}$. Specifically, $\hat{\mathbf{p}}_t(G | \hat{\boldsymbol{y}}_{1:t-1})$ means $\mathbb{P}_{y \sim \hat{\mathbf{p}}_t(\cdot | \boldsymbol{x}, \hat{\boldsymbol{y}}_{1:t-1})}[y \in G]$, parameterized by a fixed green list $G$. Similarly, $\mathbf{p}_t(G | \boldsymbol{y}_{1:t-1})$ denotes $\mathbb{P}_{y \sim \mathbf{p}_t(\cdot | \boldsymbol{x}, \boldsymbol{y}_{1:t-1})}[y \in G]$.

The proof of Theorem C.13 considers the following decomposition

$$|\boldsymbol{y}|_G = |\boldsymbol{y}|_G - \sum_t \hat{\mathbf{p}}_t(G | \hat{\boldsymbol{y}}_{1:t-1}) \tag{5}$$

$$+ \sum_t \hat{\mathbf{p}}_t(G | \hat{\boldsymbol{y}}_{1:t-1}) - \sum_t \hat{\mathbf{p}}_t(G | \boldsymbol{y}_{1:t-1}) \tag{6}$$

$$+ \sum_t \hat{\mathbf{p}}_t(G | \boldsymbol{y}_{1:t-1}) \tag{7}$$

steps to prove a lower bound to each of the three terms. We will start with the high probability bound (the second statement in Theorem C.13) then deal with the expectation.

### C.6.1 MANY GREEN LIST TOKENS WITH HIGH PROBABILITY

To obtain a high-probability lower bound, it requires us to obtain concentration for each of the three terms. Specifically,

1. To bound Term (5), we use Lemma C.16 which invokes Martingale concentration over the randomness in $\boldsymbol{y}$ to show $|\boldsymbol{y}|_G$ is close to $\sum_t \hat{\mathbf{p}}_t(G|\hat{\boldsymbol{y}}_{1:t-1})$.

2. We will show Term (6) is non-negative with high probability by using the homophily assumption (Assumption C.12). This allows us to study the roll-out $\hat{\boldsymbol{y}}_{1:t-1}$ under $\hat{\mathcal{M}}(\boldsymbol{x})$ (or $\hat{\mathbf{p}}$) by studying a hypothetical alternative roll-out $\boldsymbol{y}_{1:t-1}$ sampled under $\mathcal{M}(\boldsymbol{x})$ (or $\mathbf{p}$).

3. Then we control Term (7) by first Taylor expanding it into quantities involving $\mathbf{p}_t(G|\boldsymbol{y}_{1:t-1})$ instead of $\hat{\mathbf{p}}(G|\boldsymbol{y}_{1:t-1})$, then apply concentration inequalities for each expanded terms over the randomness of $G$ (while fixing $\boldsymbol{y}_{1:t-1}$) to obtain a high probability lower bound. Proposition C.19 gives the results.

We start by tackling (5) via Martingale concentration.

**Lemma C.16.** *For any green list $G$ and prompt $\boldsymbol{x}$.*

$$\mathbb{E}\left[|\boldsymbol{y}|_G - \sum_{t=1}^n \mathbb{P}_{y_t \sim \hat{\mathbf{p}}(\cdot|\boldsymbol{x}, \boldsymbol{y}_{1:t-1})}[y_t \in G]\right] = 0.$$

*Moreover, with probability at least $1 - \beta$ over the roll-out*

$$|\boldsymbol{y}|_G \geq \sum_{t=1}^n \mathbb{P}_{y_t \sim \hat{\mathbf{p}}(\cdot|\boldsymbol{x}, \boldsymbol{y}_{1:t-1})}[y_t \in G] - \sqrt{2n\log(2/\beta)}.$$

*Proof.* We fix $G$ and construct a martingale sequence $X_1, X_2, ..., X_n$ where $X_0 = 0$ and:

$$X_t = X_{t-1} + \mathbf{1}(y_t \in G) - \mathbb{P}_{y_t \sim \hat{\mathbf{p}}(\cdot|\boldsymbol{x}, \boldsymbol{y}_{1:t-1})}[y_t \in G].$$

Check that $\mathbb{E}[X_t|\boldsymbol{y}_{1:t-1}] = X_{t-1}$. The underlying filtration is the sigma-field generated by $y_{1:t}$.

The claim about the expectation follows from that $X_0 = 0$ and an inductive argument following the tower property of conditional probabilities.

By the fact that $|X_t - X_{t-1}| \leq 1$ we can apply Azuma-Hoeffding's inequality and get

$$\mathbb{P}\left[|X_n - \mathbb{E}[X_n]| \geq u\right] \leq 2e^{-\frac{u^2}{2n}}.$$

Check that by an inductive argument $\mathbb{E}[X_n] = 0$. So we get that with probability at least $1 - \delta$

$$|X_n| = \left|\sum_{t=1}^n \mathbf{1}(y_t \in G) - \sum_{t=1}^n \mathbb{P}_{y_t \sim \hat{\mathbf{p}}(\cdot|\boldsymbol{x}, \boldsymbol{y}_{1:t-1})}[y_t \in G]\right| \leq \sqrt{2n\log(2/\delta)}.$$

$\square$

To handle (6), we apply Assumption C.12 with parameter $\beta/3$, which says that with probability $1 - \beta/3$ (6)$\geq 0$. This converts a roll-out from $\hat{y} \sim \hat{\mathbf{p}}$ to a roll-out from the original $p$.

Before we deal with (7), let us write a lemma that rewrites $\hat{\mathbf{p}}_t(G|\boldsymbol{y}_{1:t-1})$ into a more convenient form.

**Lemma C.17.** *For any $t$, $\boldsymbol{h}_t$. Fix $G$. Denote short hands $\hat{\mathbf{p}}(G) := \mathbb{P}_{y_t \sim \hat{\mathbf{p}}_t(\cdot|\boldsymbol{x}, \boldsymbol{h}_t)}[y_t \in G]$ and $\mathbf{p}(G) := \mathbb{P}_{y_t \sim \mathbf{p}_t(\cdot|\boldsymbol{x}, \boldsymbol{h}_t)}[y_t \in G]$.*

$$\hat{\mathbf{p}}(G) = \frac{e^\delta \mathbf{p}(G)}{1 + (e^\delta - 1)\mathbf{p}(G)} = \left(1 + \frac{(e^\delta - 1)(1 - \mathbf{p}(G))}{1 + (e^\delta - 1)\mathbf{p}(G)}\right)\mathbf{p}(G).$$

*Proof.* By definition,

$$\hat{\mathbf{p}}(G) = \frac{\sum_{y \in G} e^{\ell_y + \delta}}{\sum_{y \in G} e^{\ell_y + \delta} + \sum_{y \notin G} e^{\ell_y}}$$

$$= \frac{e^\delta \mathbf{p}(G)}{e^\delta \mathbf{p}(G) + 1 - \mathbf{p}(G)} = \frac{e^\delta}{1 + (e^\delta - 1)\mathbf{p}(G)} \mathbf{p}(G)$$

$$= \left(1 + \frac{(e^\delta - 1)(1 - \mathbf{p}(G))}{1 + (e^\delta - 1)\mathbf{p}(G)}\right) \mathbf{p}(G).$$

$\square$

The lemma implies that $\hat{\mathbf{p}}(G) \geq \mathbf{p}(G)$ and that if $\mathbf{p}(G)$ is bounded away from 1, $\hat{\mathbf{p}}(G) \geq (1 + O(\delta))\mathbf{p}(G)$.

**Lemma C.18.** *For any $t$, $\mathbf{h}_t$. Fix $G$.*

$$\hat{\mathbf{p}}(G) \geq \frac{e^\delta \gamma}{1 + (e^\delta - 1)\gamma} + \frac{e^\delta}{(1 + (e^\delta - 1)\gamma)^2}(\mathbf{p}(G) - \gamma) - e^\delta(\mathbf{p}(G) - \gamma)^2$$

*Proof.* By the second-order Taylor's theorem

$$\frac{e^\delta x}{1 + (e^\delta - 1)x} = \frac{e^\delta \gamma}{1 + (e^\delta - 1)\gamma} + \frac{e^\delta}{(1 + (e^\delta - 1)\gamma)^2}(x - \gamma) - \frac{e^\delta}{(1 + (e^\delta - 1)\tilde{x})^3}(x - \gamma)^2$$

where $\tilde{x} \in [x, \gamma]$ is a function of $x$. By relaxing $\tilde{x}$ to 0 we obtain the lower bound as claimed. $\square$

Now we are ready to handle (7) with high probability in the following proposition.

**Proposition C.19** (Concentration). *For any fixed sequence $\mathbf{y}_{1:n}$, and the corresponding language model's probability distribution $\mathbf{p}$ that gives conditional distributions $\mathbf{p}_1, ..., \mathbf{p}_n$. There exists a parameter $C_{\delta,\gamma}$ that depends only $\delta, \gamma$. Then with probability at least $1 - \beta$ for any $\beta > 0$ (over $G$),*

$$\sum_{t=1}^n \mathbb{P}_{y_t \sim \mathbf{p}(\cdot | \boldsymbol{x}, \boldsymbol{y}_{1:t-1})} [y_t \in G] \geq \frac{n\gamma e^\delta}{1 + (e^\delta - 1)\gamma}$$

$$- C_{\delta,\gamma} \log^2 \frac{9(n+1)}{\beta} \left( \| \sum_{t=1}^n \mathbf{p}_t[\cdot] \| + \sum_{t=1}^n \|\mathbf{p}_t[\cdot]\|^2 + \| \sum_{t=1}^n \mathbf{p}_t[\cdot] \|_\infty + \sum_{t=1}^n \|\mathbf{p}_t[\cdot]\|_\infty^2 \right).$$

*Proof.* By Lemma C.17 and C.18

$$\sum_{t=1}^n \mathbb{P}_{y_t \sim \hat{\mathbf{p}}(\cdot | \boldsymbol{x}, \boldsymbol{y}_{1:t-1})} [y_t \in G]$$

$$= \sum_t \frac{e^\delta \mathbf{p}_t(G)}{1 + (e^\delta - 1)\mathbf{p}_t(G)}$$

$$\geq \sum_t \frac{e^\delta \gamma}{1 + (e^\delta - 1)\gamma} + \frac{e^\delta(\mathbf{p}_t(G) - \gamma)}{(1 + (e^\delta - 1)\gamma)^2} - e^\delta(\mathbf{p}_t(G) - \gamma)^2$$

$$= \frac{n\gamma e^\delta}{1 + (e^\delta - 1)\gamma} + \frac{e^\delta}{(1 + (e^\delta - 1)\gamma)^2} \underbrace{\left( \sum_t \sum_{i=1}^{N\gamma} \mathbf{p}_t[\pi[i]] - n\gamma \right)}_{(*)} - e^\delta \sum_t \underbrace{\left( \sum_{i=1}^{N\gamma} \mathbf{p}_t[\pi[i]] - \gamma \right)^2}_{(**)}$$

where $\pi$ is a random permutation of the index set $\{1, ..., N\}$.

We will now apply Lemma F.1 to lowerbound $(*)$ with high probability and to bound the absolute value of $(**)$ with high probability.

*Remark* C.20. The reason why we can apply these lemmas even after we condition on $\boldsymbol{y}_{1:t-1}$ is due to the "high-probability homophily" assumption which allows us to use the fact that $\boldsymbol{y}_{1:t-1}$ is independent to $G$, i.e., the distribution of the green list remains uniform at random after we condition on each qualifying $\boldsymbol{y}_{1:t-1}$ separately.

Using a similar argument from the proof of Theorem C.4, we can apply Lemma F.1 and get that with probability $1 - \beta$,

$$(*) \geq -\sqrt{64\gamma \| \sum_{t=1}^{n} \mathbf{p}_t(\cdot) \|^2 \log(9/\beta)} - \| \sum_{t=1}^{n} \mathbf{p}_t(\cdot) \|_\infty \log(9/\beta).$$

Similarly by Lemma F.1 again to bound $(**) = \sum_{i=1}^{N\gamma} \mathbf{p}_t[\pi[i]] - \gamma$ w.h.p for each $t$.

$$\big|(**)\big| \leq \sqrt{64\gamma \|\mathbf{p}_t(\cdot)\|^2 \log(9/\beta)} + \|\mathbf{p}_t(\cdot)\|_\infty \log(9/\beta).$$

To put things together, with probability $1 - (n+1)\beta$,

$$\sum_{t=1}^{n} \mathbb{P}_{y_t \sim \mathbf{p}(\cdot|\boldsymbol{x}, \boldsymbol{y}_{1:t-1})} [y_t \in G]$$

$$\geq \frac{n\gamma e^\delta}{1 + (e^\delta - 1)\gamma} - \frac{e^\delta}{(1 + (e^\delta - 1)\gamma)^2} \left( \sqrt{64\gamma \| \sum_{t=1}^{n} \mathbf{p}_t[\cdot] \|^2 \log(9/\beta)} + \| \sum_{t=1}^{n} \mathbf{p}_t[\cdot] \|_\infty \log(9/\beta) \right)$$

$$- e^\delta \gamma(1 - \gamma) \sum_t \|\mathbf{p}_t[\cdot]\|^2 - 2e^\delta \left( 64\gamma \sum_{t=1}^{n} \|\mathbf{p}_t[\cdot]\|_2^2 \log(9/\beta) + \sum_{t=1}^{n} \|\mathbf{p}_t[\cdot]\|_\infty^2 \log^2(9/\beta) \right)$$

$$\geq \frac{n\gamma e^\delta}{1 + (e^\delta - 1)\gamma} - C_{\delta,\gamma} \log(9/\beta)^2 \left( \| \sum_{t=1}^{n} \mathbf{p}_t[\cdot] \| + \sum_{t=1}^{n} \|\mathbf{p}_t[\cdot]\|^2 + \| \sum_{t=1}^{n} \mathbf{p}_t[\cdot] \|_\infty + \sum_{t=1}^{n} \|\mathbf{p}_t[\cdot]\|_\infty^2 \right)$$

for a constant $C_{\delta,\gamma}$ that depends only in $\delta, \gamma$. The proof is complete by defining $\tilde{\beta} = 9(n+1)\beta$, and get the same result under probability $1 - \tilde{\beta}$. $\qquad \square$

### C.6.2 MANY GREEN LIST TOKENS IN EXPECTATION

To obtain the lower bound in expectation, we just need to bound the expectation of (5), (6) and (7).

1. Observe that $\mathbb{E}[\text{Term } (5)|G] = 0$ (from Lemma C.16)
2. Also, observe that $(6) \geq 0$ under the *homophily* assumption (Assumption C.11).
3. Term (7) can be further lower bounded by a second-order Taylor expansion argument (Lemma C.18) and a variance calculation for sampling without replacement (Lemma C.21), which ends up depending on the *on-average high-entropy* parameter from Definition C.8. The formal result is stated in Proposition C.22.

**Lemma C.21.** *Fix* $\mathbf{p}_t$

$$\mathbb{E}_G[(\mathbf{p}_t(G) - \gamma)^2] \leq \gamma(1 - \gamma)\|\mathbf{p}_t[\cdot]\|^2.$$

*Proof.* First observe that $\mathbb{E}_G[\mathbf{p}_t(G)] = \gamma$ because every token has $\gamma$ probability to be included. By the variance formula for sampling without replacement ($N$ choose $N\gamma$),

$$\text{Var}_G[\mathbf{p}_t(G)|\boldsymbol{y}_{1:t-1}] = \gamma N \frac{1}{N} \sum_{i=1}^{N} (\mathbf{p}_t[i]^2 - N^{-2})(1 - \frac{\gamma N - 1}{N - 1}) \leq \gamma(1 - \gamma) \sum_{i=1}^{N} \mathbf{p}_t[i]^2.$$

$\square$

**Proposition C.22.** *Assume homophily, then*

$$\mathbb{E}\left[ \sum_{t=1}^{n} \mathbb{P}_{y_t \sim \hat{\mathbf{p}}(\cdot|\boldsymbol{x}, \boldsymbol{y}_{1:t-1})} [y_t \in G] \right] \geq n\gamma \left( \frac{e^\delta}{1 + (e^\delta - 1)\gamma} - \frac{(1-\gamma)e^\delta}{n} \sum_{t=1}^{n} \mathbb{E}_{\boldsymbol{y}_{1:t-1} \sim \mathbf{p}(\cdot|\boldsymbol{x})} \sum_{i=1}^{N} \mathbf{p}_t[i]^2 \right).$$

*Proof.* By homophily,

$$\mathbb{E}\left[\sum_{t=1}^{n} \mathbb{P}_{y_t \sim \hat{\mathbf{p}}(\cdot|\boldsymbol{x}, \boldsymbol{y}_{1:t-1})}[y_t \in G]\right]$$

$$=\sum_{t=1}^{n} \mathbb{E}_{G, \boldsymbol{y}_{1:t-1} \sim \hat{\mathbf{p}}(\cdot|\boldsymbol{x})}\left[\mathbb{P}_{y_t \sim \hat{\mathbf{p}}(\cdot|\boldsymbol{x}, \boldsymbol{y}_{1:t-1})}[y_t \in G]\right]$$

$$\geq \sum_{t=1}^{n} \mathbb{E}_{G, \boldsymbol{y}_{1:t-1} \sim \mathbf{p}(\cdot|\boldsymbol{x})}\left[\mathbb{P}_{y_t \sim \hat{\mathbf{p}}(\cdot|\boldsymbol{x}, \boldsymbol{y}_{1:t-1})}[y_t \in G]\right]$$

$$=\sum_{t=1}^{n} \mathbb{E}_{\boldsymbol{y}_{1:t-1} \sim \mathbf{p}(\cdot|\boldsymbol{x})} \mathbb{E}_{G}\left[\frac{e^\delta \, \mathbb{P}_{y_t \sim \mathbf{p}_t(\cdot|\boldsymbol{y}_{1:t-1})}[y_t \in G]}{1 + (e^\delta - 1)\, \mathbb{P}_{y_t \sim \mathbf{p}_t(\cdot|\boldsymbol{y}_{1:t-1})}[y_t \in G]}\bigg| \boldsymbol{y}_{1:t-1}\right] \tag{8}$$

By Lemma C.18, we can decompose (8). Also observe that $\mathbb{E}_G\left[\mathbf{p}_t(G)\big|\boldsymbol{y}_{1:t-1}\right] = \gamma$ where $\mathbf{p}_t(G) :=$ $\mathbb{P}_{y_t \sim \mathbf{p}_t(\cdot|\boldsymbol{y}_{1:t-1})}[y_t \in G]$ is short hand for clarity. To see the second observation, notice that $y_t$ is independent to $G$, thus we can apply Statement 1 of Theorem C.4.

Apply the two observations to (8), we have

$$(8) \geq \sum_{t=1}^{n} \mathbb{E}_{\boldsymbol{y}_{1:t-1} \sim \mathbf{p}(\cdot|\boldsymbol{x})} \mathbb{E}_{G}\left[\frac{e^\delta \gamma}{1+(e^\delta-1)\gamma} + \frac{e^\delta(\mathbf{p}_t(G) - \gamma)}{(1+(e^\delta-1)\gamma)^2} - e^\delta(\mathbf{p}_t(G) - \gamma)^2\bigg|\boldsymbol{y}_{1:t-1}\right]$$

$$=\frac{e^\delta n\gamma}{1+(e^\delta-1)\gamma} + \sum_{t=1}^{n} \mathbb{E}_{\boldsymbol{y}_{1:t-1} \sim \mathbf{p}(\cdot|\boldsymbol{x})}\left[\frac{e^\delta(\mathbb{E}_G[\mathbf{p}_t(G)|\boldsymbol{y}_{1:t-1}] - \gamma)}{(1+(e^\delta-1)\gamma)^2} - e^\delta \mathbb{E}_G[(\mathbf{p}_t(G) - \gamma)^2|\boldsymbol{y}_{1:t-1}]\right]$$

$$=\frac{e^\delta n\gamma}{1+(e^\delta-1)\gamma} - \sum_{t=1}^{n} e^\delta \mathbb{E}_{\boldsymbol{y}_{1:t-1} \sim \mathbf{p}(\cdot|\boldsymbol{x})} \mathrm{Var}_G[\mathbf{p}_t(G)|\boldsymbol{y}_{1:t-1}].$$

By the variance formula for sampling without replacement ($N$ choose $N\gamma$),

$$\mathrm{Var}_G[\mathbf{p}_t(G)|\boldsymbol{y}_{1:t-1}] = \gamma N \frac{1}{N} \sum_{i=1}^{N} (\mathbf{p}_t[i]^2 - N^{-2})(1 - \frac{\gamma N - 1}{N-1}) \leq \gamma(1-\gamma) \sum_{i=1}^{N} \mathbf{p}_t[i]^2.$$

Thus it follows that

$$(8) \geq \frac{e^\delta n\gamma}{1+(e^\delta-1)\gamma} - \sum_{t=1}^{n} e^\delta \mathbb{E}_{\boldsymbol{y}_{1:t-1} \sim \mathbf{p}(\cdot|\boldsymbol{x})} \gamma(1-\gamma) \sum_{i=1}^{N} \mathbf{p}_t[i]^2$$

$$=n\gamma\left(\frac{e^\delta}{1+(e^\delta-1)\gamma} - \frac{(1-\gamma)e^\delta}{n} \sum_{t=1}^{n} \mathbb{E}_{\boldsymbol{y}_{1:t-1} \sim \mathbf{p}(\cdot|\boldsymbol{x})} \sum_{i=1}^{N} \mathbf{p}_t[i]^2\right).$$

$\square$

## C.7 SECURITY PROPERTY

**Corollary C.23.** *Algorithm 2 with threshold $\tau$ satisfies the **security property** from Definition 2.2 with $\epsilon = 0$ and*

$$\eta(\boldsymbol{y}, \mathsf{k}, \epsilon) = \frac{\sqrt{n}(z_{\boldsymbol{y}} - \tau)}{1 + \gamma/2} \mathbf{1}\left(z_{\boldsymbol{y}} - \tau \geq \frac{\gamma\sqrt{n}}{1+\gamma/2}\right).$$

In comparison, the best bound on the security property parameter one can obtain for the scheme of Kirchenbauer et al. (2023) is (a formal statement and proof are included in Appendix D.2)

$$\eta(\boldsymbol{y}, \mathsf{k}, \epsilon) = \frac{\sqrt{n}(z_{\boldsymbol{y}} - \tau)}{2 + \gamma/2} \mathbf{1}\left(z_{\boldsymbol{y}} - \tau \geq \frac{\gamma\sqrt{n}}{2+\gamma/2}\right).$$

To say it differently, our method, UNIGRAM-WATERMARK, utilizing a fixed Green-Red split, achieves *twice the robustness* to edits compared to Kirchenbauer et al. (2023)'s baseline approach.

## D    ANALYSIS OF KIRCHENBAUER ET AL. (2023)

### D.1    SOFT WATERMARKING SCHEME OF KIRCHENBAUER ET AL. (2023)

This section illustrates the soft watermarking scheme proposed by Kirchenbauer et al. (2023). This straightforward algorithm only requires access to the language model's logits at each time step. Let $\boldsymbol{y} = [y_1, \ldots, y_n]$ represent the output sentence of language model $\mathcal{M}$ given the prompt $\boldsymbol{x}$. The watermarking scheme generates $\boldsymbol{y}_{1:n}$ by hashing $y_{t-1}$ to a partition of the token space (Green List and Red List) and amplifies the probability of tokens on the Green List. Specifically, $[y_1, \ldots, y_n]$ is derived from the following Markov chain:

1. $y_1 \sim \text{Softmax}\big(\text{logits}_{\mathcal{M}}\big(y_1 = \cdot|x\big)\big)$

2. For $t = 2 : n$,

$$y_t \sim \text{Softmax}\big(\text{logits}_{\mathcal{M}}(y_t = \cdot|[\boldsymbol{x}, y_1 \ldots, y_{t-1}]) + \delta\mathbf{1}(\cdot \in \text{Green}(y_{t-1}))\big)$$

Typically, $\gamma|\mathcal{V}|$ tokens are selected to form a Green List, where $\gamma$ symbolizes the fraction of tokens to be watermarked (by default, $\gamma = 0.5$). The logit value for each green token is augmented by a constant $\delta$ (default value $= 2$), which denotes the watermark strength. This elevation enhances the likelihood of sampling green, watermarked tokens, particularly for high-entropy distributions.

Validation of whether a text was generated by a watermarked language model is achievable given knowledge of the hash function and tokenizer. The adversary constructs $\boldsymbol{u} = [u_1, \ldots, u_m]$ from $\boldsymbol{x}, \boldsymbol{y}_{1:n}$ and any auxiliary input. The detection algorithm calculates the quantity of green tokens $|\boldsymbol{u}|_G = \sum_{t=2}^{m} \mathbf{1}(u_t \in \text{Green}(u_{t-1}))$. One can assume the null hypothesis, denoted as $H_0$: *The text sequence is produced independently of the green list rule*. Following this, a $z$-statistic score is computed as $z = (|\boldsymbol{u}|_G - \gamma m) / \sqrt{m\gamma(1-\gamma)}$. If the $z$-score exceeds a predetermined threshold, the algorithm declares, "This was generated from $\hat{\mathcal{M}}$!".

### D.2    SECURITY PROPERTY OF KIRCHENBAUER ET AL. (2023)

We also demonstrate the robustness property of the soft watermarking algorithm in Kirchenbauer et al. (2023) in the following Theorem D.1

**Theorem D.1** (Robustness to editing in the watermarking scheme of Kirchenbauer et al. (2023))**.** *Let $\boldsymbol{y} = [y_1, \ldots, y_n]$ represent the watermarked sequence. Suppose the adversary $\mathcal{A}$ follows the definition 2.2 and outputs a modified text $\boldsymbol{u} = [u_1, \ldots, u_m]$. Following Equation 2, we calculate the z-score of the soft watermarking Kirchenbauer et al. (2023) $z_{\boldsymbol{y}}$ and $z_{\boldsymbol{u}}$. Then we have*

$$z_{\boldsymbol{u}} \geq z_{\boldsymbol{y}} - \max\{\frac{(2+\gamma/2)\eta}{\sqrt{n}}, \frac{(2-\gamma/2)\eta}{\sqrt{n-\eta}}\}.$$

*Proof.* The proof is similar to that of Theorem 3.7 except that the maximum perturbation to $|\mathbf{y}|_G$ is now $2\eta$ rather than $\eta$. We now justify that the maximum perturbation has really doubled below, but ignore the part that is the same as in the proof of Theorem 3.7.

Let $\text{BiGrams}(\boldsymbol{u}) = \{\{u_1, u_2\}, \{u_2, u_3\}, ..., \{u_{n-1}, u_n\}\}$ and similarly $\text{BiGrams}(\boldsymbol{y})$ enumerates the set of all two grams in sequence $\boldsymbol{y}_{1:m}$.

We claim that each edit can modify at most two elements in the above set. To see this, consider "insertion", "deletion", and "edit" separately.

- If we "insert" one token $\tilde{u}$ at $t$, then $\{u_{t-1}, u_t\}$ and $\{u_t, u_{t+1}\}$ become $\{u_{t-1}, \tilde{u}\}$, $\{\tilde{u}, u_t\}$ and $\{u_t, u_{t+1}\}$. Only one element of $\text{BiGrams}(\boldsymbol{u})$ is modified — $\{u_{t-1}, u_t\}$.

- For "deletion" at $t$, $\{u_{t-1}, u_t\}$ and $\{u_t, u_{t+1}\}$ become $\{u_{t-1}, u_{t+1}\}$. So two elements from $\text{BiGrams}(\boldsymbol{u})$ are gone.

- For "edit" at $t$, $\{u_{t-1}, u_t\}$ and $\{u_t, u_{t+1}\}$ become $\{u_{t-1}, \tilde{u}\}$ and $\{\tilde{u}, u_{t+1}\}$. Thus again only two elements from $\text{BiGrams}(\boldsymbol{u})$ are gone.

It follows that when $\boldsymbol{y}$ is obtained after up to $\eta$ edits

$$|\mathsf{BiGrams}(\boldsymbol{u}) \cap \mathsf{BiGrams}(\boldsymbol{y})| \geq |\mathsf{BiGrams}(\boldsymbol{u})| - 2\eta$$

Observe that $\sum_{t=2}^{n} \mathbf{1}(u_t \in \mathsf{Green}(u_{t-1}))$ counts the number of qualifying elements in $\mathsf{BiGrams}(\boldsymbol{u})$, which completes the proof. $\qquad \square$

**For this reason, our watermark is twice as robust as that of Kirchenbauer et al. (2023). This provides the theoretical guarantee to our empirical results presented in the experiments!**

*Remark* D.2. We can view our watermark as a trivial Markovian watermarking scheme with $k = 0$, and what Kirchenbauer et al. (2023) proposed to be $k = 1$. For the more general $k$-Markovian watermarking scheme that depends on a prefix of length $k$, the robustness deteriorates by a factor of $k$, as the maximum perturbation will become $\frac{((k+1)+\gamma/2)\eta}{\sqrt{n}}$. To say it differently, choosing $k = 0$ gives the maximum robustness and maximum simplicity at the same time, and the benefit leads to significant gains in our experiments, especially against paraphrasing attacks.

# E  ALTERNATIVE DETECTOR "UNIQUE" AND ITS DESIRABLE PROPERTIES

Our theoretical analysis suggests a promising alternative Detect algorithm for UNIGRAM-WATERMARK that simply involves calling Algorithm 2 with a deduplicated $\boldsymbol{y}$.

---

**Algorithm 4** UNIGRAM-WATERMARK: Detect (Alternative)

---

1: **Input:** suspect text $\boldsymbol{y}$, watermark detection key k, threshold $\tau$.
2: **Output:** 1 or 0 (whether the text is watermarked).
3: **Return** Algorithm 2 with suspect text $\mathrm{Unique}(\boldsymbol{y})$, detection key k and threshold $\tau$.

---

The simple change actually results in a number of interesting new properties. For example, we can state its Type I error bound a lot more cleanly now as a Corollary of Theorem C.4

**Corollary E.1** (No false positive for Deduplicated Detection). *Consider $\boldsymbol{y} = \boldsymbol{y}_{1:n}$ as any fixed suspect text. Let $m = |\mathrm{Unique}(\boldsymbol{y})|$ be the number of unique tokens in $\boldsymbol{y}$. Let $G$ be selected through Algorithm 1, using a uniform random choice. Then the following statements hold true:*

*1. Assume $m \geq 1$, then*

$$\mathbb{E}[|\mathrm{Unique}(\boldsymbol{y})|_G | \boldsymbol{y}] = \gamma n \quad \text{and} \quad \mathbb{E}[z_{\mathrm{Unique}(\boldsymbol{y})} | \boldsymbol{y}] = 0.$$

*2. With probability $1 - \alpha$ (over only the randomness of $G$),*

$$\mathbb{P}\left[|\mathrm{Unique}(\boldsymbol{y})|_G \geq \gamma m + \sqrt{64\gamma m \log(9/\alpha)} + \log(9/\alpha) \Big| \boldsymbol{y}\right] \leq \alpha$$

*or equivalently (when $n \geq 1$)*

$$\mathbb{P}\left[z_{\mathrm{Unique}(\boldsymbol{y})} \geq \sqrt{\frac{64\log(9/\alpha)}{(1-\gamma)}} + \frac{\log(9/\alpha)}{\sqrt{m\gamma(1-\gamma)}} \Big| \boldsymbol{y}\right] \leq \alpha.$$

The above gives a clean finite-sample concentration bound of the Type I error using Algorithm 4. Notably, while deduplicating reduces the length of the suspect text, i.e., $m < n$, it improves the bound by ensuring both $C_{\max}$ and $V$ are 1.

*Remark* E.2 (Asymptotic choice of $\tau$ for controlling false positives). Lemma C.21 gives that

$$\mathrm{Var}\left[|\mathrm{Unique}(\boldsymbol{y})|_G | \boldsymbol{y}\right] = m\gamma(1-\gamma)(1 - \frac{m-1}{N-1})$$

i.e., the conditional variance of $z_{\mathrm{Unique}(\boldsymbol{y})}$ is $(1 - \frac{m-1}{N-1})$. This means that if we want to control the asymptotic false positive rate to $\alpha$, all we have to do is to choose the threshold $\tau$ to be

$$\tau = \sqrt{1 - \frac{m-1}{N-1}} \Phi^{-1}(1-\alpha) \tag{9}$$

where $\Phi$ is the standard normal CDF.

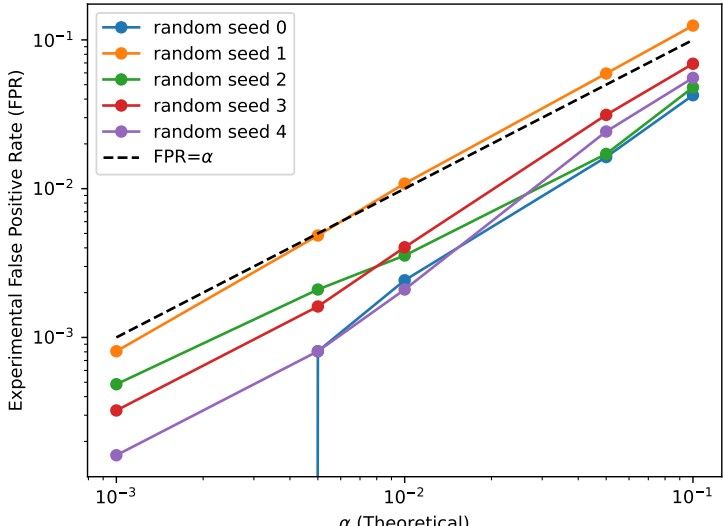

Figure 5: Empirical vs. theoretical false positive rates across various $\alpha$ values, using multiple green list initializations.

**Type II error.** How about Type II error? Our results in Theorem C.13 are still applicable but require us to apply that with a special language model derived from the original that directly generates Unique($\boldsymbol{y}$) (ordered in the same order they appear in $\boldsymbol{y}$). This is still a valid autoregressive language model but has different roll-out probabilities.

**Robustness to Edits.** Observe that adding/removing/replacing one token to $\boldsymbol{y}$ in the results in adding/removing/replacing one token to Unique($\boldsymbol{y}$) respectively, the robustness of the $z$-score for Unique($\boldsymbol{y}$) thus directly follows Theorem D.1.

**"Unique" in $K$-gram watermark section with $K \geq 2$.** Clearly, the same idea of deduplication works for the whole family of $K$-gram watermark proposed in Kirchenbauer et al. (2023). In fact, it was briefly mentioned in a remark from their paper as a mitigation measure to reduce correlation. All arguments we make about Type I error and Robustness to Edits above work for $K \geq 2$. We defer the Type II error bound for this family to a longer version of the paper.

**Emperical analysis on controlling false positives.** We conduct experiments to demonstrate the results for the asymptotic choice of $\tau$ in controlling false positives. The negative examples are sampled from diverse datasets, including human data in LFQA and OpenGen dataset (Krishna et al., 2023), C4 dataset (Raffel et al., 2020), and TOEFL dataset (Liang et al., 2023). In total, we collect 6,200 unwatermarked text samples with varied lengths. We then use the dynamic threshold $\tau$ with different choices of $\alpha$ as shown in Equation 9. By choosing different random seeds, we obtain different green lists. The results in Figure 5 show the empirical false positive rate aligns well with the theoretical $\alpha$.

### E.1 ALTERNATIVE DETECTION "UNIQUE" IS ROBUST TO EMOJI ATTACK AND OTHER TRICKY ATTACKS

Kirchenbauer et al. (2023) discussed a number of interesting attacks on the $K$-gram watermarks. In this section, we inspect the robustness of UNIGRAM-WATERMARK (with both Algorithm 2 and 4 as Detect) to these attacks.

We will focus on those trickier generative attacks, as those non-generative attacks on the surface level (e.g., synonym substitution, Unicode substitution) were rather satisfactorily addressed in Kirchenbauer et al. (2023). The same arguments work for UNIGRAM-WATERMARK. However, there are trickier ones that break $K$-gram watermarks for $K \geq 2$ but not for $K = 1$, especially when we use Algorithm 4 for detection.

**Emoji attack** the Emoji attack, also known as the Pineapple attack, asks the language model to inject a special symbol, e.g., an Emoji, in between the actual text that the LM is supposed to generate in response to a prompt. For example, a user of the language model can prompt an LM with "`Write my college admission essay. Insert an emoji in between every word.`". Then the user can simply remove the artificially injected symbol before submitting the essay. Clearly, this attack breaks all $K$-gram watermarks for $K > 1$ (Kirchenbauer et al., 2023) (but also (Aaronson, 2023)). Our UNIGRAM-WATERMARK remains effective because half of the tokens in $\boldsymbol{y}$ are still watermarked. The repeated Emoji symbol with length $n/2$ is a problem for both Type I and Type II error, but after "Unique", this corresponds to just an Edit Distance of 1 on $\mathrm{Unique}(\boldsymbol{y})$!

**Alphabet attack** We can also make the Emoji attack stronger by injecting a known sequence of "alphabets" instead of one single symbol. For example, "`Write my college admission essay. Insert, in the order of the vocabulary, a token in between every word you generate.`". The alphabet attack breaks $K$-gram watermarks in the same way an Emoji attack does, but since the added tokens are different, "Unique" does not solve it right away. Interestingly, despite $m/2$ of the tokens in $\mathrm{Unique}(\boldsymbol{y})$ are not watermarked, as long as the Alphabet being used is independent of the secret key, it does not change the Type I error and only slightly reduces the power (i.e., 1-Type II error) since the expected number of Green tokens in that $m/2$ injected tokens is $m\gamma/2$.

**Stegnography attack** One may extend the attack even further by asking the language model to encode a message, which swaps each token in the vocabulary with another token through a secret codebook. For example, whenever you want to output Token $i$, output Token $\mathrm{mod}(i + 1, N)$ instead. If the "code book" is supplied in the prompt with an instruction for the LM to follow the code book when generating the text, then it really breaks all watermarks including ours, while allowing the user who knows the code book to easily revert it to the original text. The issue of such an attack is that it requires significantly heavy-lifting for the language model to predict outside the typical distribution it is trained on. There is no real risk of such an attack being employed as it is likely to significantly reduce the quality of the generated text.

To be clear, these attacks are, in fact, not post-processing-based evasion attacks, but rather hacks into prompts. Nevertheless, our watermark that is robust to edits turns out to be quite resilient to them.

## F    TECHNICAL LEMMAS

**Lemma F.1** (Bernstein-style inequality for random permutation (Albert, 2019, Proposition 2.2)). *Let $\{a_{i,j}\}_{1 \leq i,j \leq n}$ be a collection of non-negative numbers and $\Pi_n$ be a random uniform permutation. Let $Z_n = \sum_{i=1}^{n} a_{i,\Pi_n(i)}$. Then, for any $t > 0$*

$$\mathbb{P}\left[|Z_n - \mathbb{E}[Z_n]| \geq 2\sqrt{\frac{t}{n}\sum_{i,j=1}^{n} a_{i,j}^2} + \max_{1 \leq i,j \leq n}\{a_{i,j}\}t\right] \leq 8e^{1/16}e^{-\frac{t}{16}}.$$

**Lemma F.2** (Variance for sampling without replacement). *Let $x_1, ..., x_N \in \mathbb{R}$. For any sample size $1 \leq n \leq N$, and $\pi$ be a random permutation of $\{1, 2, ..., N\}$. The variance of $X = \frac{1}{n}\sum_{i=1}^{n} x_{\pi(i)}$ satisfies*

$$\mathrm{Var}(X) = \frac{1}{nN}\sum_{i=1}^{N}(x_i - \bar{x})^2(1 - \frac{n-1}{N-1}).$$

**Definition F.3** (Martingale). A sequence of random variables $(X_n)_{n \in \mathbb{N}}$ is called a *martingale* if it satisfies the following conditions:

1. $\mathbb{E}[|X_n|] < \infty$ for all $n \in \mathbb{N}$.

2. $\mathbb{E}[X_{n+1}|\mathcal{F}_n] = X_n$ for all $n \in \mathbb{N}$.

where $\mathcal{F}_1 \subseteq \mathcal{F}_2 \subseteq ... \subseteq \mathcal{F}_n \subseteq \mathcal{F}_{n+1} \subseteq ...$ is a filtration. Specifically, $\mathcal{F}_n$ can be the sigma-algebra generated by another sequence of random variable $Y_1, ..., Y_n$, i.e., $\mathcal{F}_n = \sigma(Y_{1:n})$ and $X_n$ can be a function of $Y_{1:n}$.

**Lemma F.4** (Azuma-Hoeffding Inequality). *Let $(X_n)_{n \in \mathbb{N}}$ be a martingale such that $|X_{n+1} - X_n| \leq c_n$ for some constants $c_n$ and all $n \in \mathbb{N}$. Then for all $t > 0$ and $n \in \mathbb{N}$, we have*

$$\mathbb{P}\left(|X_n - X_0| \geq t\right) \leq 2 \exp\left(-\frac{t^2}{2 \sum_{i=1}^n c_i^2}\right).$$

