# OpenReview forum: "Provable Robust Watermarking for AI-Generated Text"
_ICLR.cc/2024/Conference — ICLR 2024 poster_

### Official Review · Reviewer_Rdg2 · 2023-10-24

**Soundness:** 3 good
**Presentation:** 3 good
**Contribution:** 3 good
**Rating:** 8
**Confidence:** 3

**Summary:**

This paper focuses on the problem of watermarking large language models generated text. The key idea is to make the logits of the language model for a specific subset of vocabulary are increased by δ.
The paper addresses three significant aspects of watermarking, including the quality of watermarked model, the performance of watermark detection algorithm, and its robustness against attacks. Firstly, the paper proves the guaranteed generation quality by “ω-Quality of watermarked output” and evaluates text perplexity of both watermarked and un-watermarked generated text.  Secondly, the paper evaluates the performance of watermark detection from two perspectives: false positive and false negatives. Lastly, the paper delves into a comprehensive analysis and presents experimental results regarding the robustness against paraphrasing attacks and editing attacks.
Overall, the research topic is meaningful and promising, the problem is well-defined, and the paper is well-written.

**Strengths:**

1.	This work is very solid. In particular, Section 2.1 stands out for its precision and conciseness, setting a solid framework for the entire paper.
2.	The paper proves Type I/Type II errors of the detection algorithm decay exponentially as the suspect text length gets longer and more diverse, which is a desirable property of watermarking generated text.
3.	The paper conducts extensive and comprehensive experiments. The proposed method achieves superior detection accuracy and improved robustness against different attacks, thus promoting the responsible use of LLMs.

**Weaknesses:**

The work is solid, and many conclusions and experimental results are provided in Appendix. My only concern is the proposed method is constrained with the length of text. Based on the key idea of the watermarking method, it can only perform well when the generated text is long enough. It is evident that longer text exhibits better performance of watermarking technique, which is indeed a favorable attribute. However, However, it raises questions about the method's viability and performance on shorter texts. Experiments testing how the proposed method performs on different text lengths are absent.

**Questions:**

1.	In the section ‘INTRODUCTION’, you mentioned that “Type I/Type II errors of the detection algorithm decay exponentially as the suspect text length gets longer and more diverse.” I'm curious about how do you assess the diversity of text in your paper.
2.	While the paper applies a constraint based on edit distance to evaluate the security property, it's worth noting that some attacks may not significantly alter the edit distance. In such cases, how do we effectively assess the security of the watermark?
3.	I'm intrigued by the observation that a smaller gamma value results in a stronger watermark. Intuitively, a larger gamma would seem to imply that more words in the vocabulary carry the watermark. However, it's also apparent that a larger gamma can lead to an increase in false positives. Could you please provide further clarification on this relationship?

---

> ### Author Response · Authors · 2023-11-20
> **Rebuttal Response to Reviewer Rdg2**
>
> Thank you for your support of our paper!
>
> ### **Impact of Text Length on Watermark Detection**
> > However, it raises questions about the method's viability and performance on shorter texts. Experiments testing how the proposed method performs on different text lengths are absent.
>
> To investigate the effect of text length on the effectiveness of our watermark detection method, we conducted experiments across different text lengths under both normal and attack scenarios. Specifically, we truncated the text to lengths of 50, 100, 150, and 200 tokens, and evaluated the detection performance. The attack is done by ChatGPT paraphrasing attack. The results, detailed below, demonstrate the robustness of our approach across varying text lengths.
>
> | Length | Setting | AUC   | TPR at 1% FPR | F1 score at 1% FPR | TPR at 10% FPR | F1 score at 10% FPR |
> |-|-|-|-|-|-|-|
> | 50     | Normal  | 0.993 | 0.892         | 0.932              | 0.987          | 0.899               |
> | 100    | Normal  | 0.999 | 0.987         | 0.987              | 1.0            | 0.911               |
> | 150    | Normal  | 1.0   | 1.0           | 0.989              | 1.0            | 0.907               |
> | 200    | Normal  | 1.0   | 1.0           | 0.995              | 1.0            | 0.952               |
>
> | Length | Setting | AUC   | TPR at 1% FPR | F1 score at 1% FPR | TPR at 10% FPR | F1 score at 10% FPR |
> |-|-|-|-|-|-|-|
> | 50     | Attack  | 0.886 | 0.297         | 0.451              | 0.696          | 0.731               |
> | 100    | Attack  | 0.948 | 0.525         | 0.683              | 0.858          | 0.835               |
> | 150    | Attack  | 0.968 | 0.661         | 0.787              | 0.915          | 0.863               |
> | 200    | Attack  | 0.985 | 0.829         | 0.896              | 0.953          | 0.883               |
>
> These results highlight the consistent and effective detection of watermarks across various text lengths, affirming the adaptability and robustness of our method. Generally, for shorter text, the detection is not perfect but still achieves very high AUC scores. As the length increases, the detection performance improves.
>
> ### **Response to Specific Questions**
>
> **Assessing Text Diversity.** In our paper, the diversity of text is primarily assessed based on the “on-average high entropy” assumption, as texts typically exhibit a wide range of vocabulary and syntactic patterns. The “on-average high entropy” assumption requires the probability distribution of the roll-out text to be “sufficiently diverse” on average. It is related to but distinct from the “spike entropy” assumption used by Kirchenbauer et al. (2023). We provide the definition and a discussion of this assumption in Appendix C.4.1.
>
> **Security Assessment Against Edit-Distance Invariant Attacks.** Our watermarking technique uses edit distance as a measure of robustness. The edit distance is the only metric we require the watermark to be robust against. If some attacks do not significantly alter the edit distance, then we can still detect the watermarked text robustly.
>
>
> **Clarification on Gamma Value and Watermark Strength.**
> The watermark strength increases as $\delta$ increases, but the dependence on \gamma is more complex. Increasing $\gamma$ does increase the number of green tokens as the reviewer observes, but it also changes the variance and the tail bound. Bear in mind that the detection rule relies not on the number of green tokens in absolute number, but how many more green tokens we observe over the natural range of the null distribution,
> It is also coupled with $\delta$, for the same choice of $\delta$, smaller $\gamma$ ensures that the selected green-tokens stand out a bit more due to the soft-max normalization. That might have explained our empirical finding in Table 5.  Finding the ideal choice of $\gamma$ theoretically and empirically is an interesting question.    At the moment, for every choice of $\gamma,\delta$, our theory allows us to choose a threshold that controls the false positive rate (Remark 3.4).

---

### Official Review · Reviewer_Po1T · 2023-10-25

**Soundness:** 3 good
**Presentation:** 2 fair
**Contribution:** 3 good
**Rating:** 8
**Confidence:** 4

**Summary:**

Built upon the watermark framework proposed in (Kirchenbauer et al., 2023), this paper studies a theoretically provable watermarking framework, named UNIGRAM-WATERMARK, with provable type-I and type-II error analysis. Empirical results showed improved performance over (Kirchenbauer et al., 2023).

**Strengths:**

1. The proposed method is theoretically sound - it provides type-I and type-II error analysis and improves the baseline line method proposed in (Kirchenbauer et al., 2023).

2. Improved watermarking performance (in terms of detection rate and utility/perplexity) over (Kirchenbauer et al., 2023). The empirical results are thorough and convincing.

3. The paper showed robustness results and analysis against (simple) text editing.

**Weaknesses:**

1. Technically, the paper is sound and novel in deriving provably watermarks. However, I have some questions on the definition of false positives (see Questions).

2. On the presentation side, it reads like the authors tend to contrast their approach with the K-gram method, and the discussion has been brought up multiple times in the paper. However, only some simple text such as "general K-gram watermark works in the same way as ours, but randomly generates a different Green list for each prefix of length K − 1. In contrast, choosing K = 1 means we have a consistent green list for every new token the language model generates.'' It would be better if the authors could provide an algorithmic description on the K-gram method so the readers can directly compare it with Algorithms 1 and 2.

**Questions:**

1. In Definition 2.2, when comparing the defined type-I and type-II errors, I am confused about why in the type-II error, we need to assume $y$ is generated from the model $\hat{M}$, whereas in the type-I error, it is assumed that $y$ can be arbitrary. Wouldn't it make more sense to assume  that $y$ is **not** generated from model $\hat{M}$ in the type-I error analysis? Based on the current definition, if $y$ is indeed generated by $\hat{M}$, then the definition says its detection correctness is at most $\alpha$? Please clarify it.

2. For the general K-gram method, can the error guarantees and analysis be extended? If so, how will this general K value place a role in the error analysis?

---

> ### Author Response · Authors · 2023-11-20
> **Rebuttal Response to Reviewer Po1T**
>
> We appreciate the reviewer's acknowledgment of our contribution.
>
> ### **Compare with the K-gram Method**
> > It would be better if the authors could provide an algorithmic description on the K-gram method so the readers can directly compare it with Algorithms 1 and 2.
>
> We appreciate your insightful suggestion. Indeed, we have included a detailed algorithmic description of Kirchenbauer et al. 2023 in Appendix D.1. The primary distinction between Kirchenbauer et al. and the K-gram method lies in the modification of the green list from Green($y_{t-1}$) to Green($y_{t-k-1:t-1}$). We acknowledge the importance of this comparison and will incorporate a detailed discussion in the revised manuscript.
>
> ### **Clarification on Type-I and Type-II Errors**
> > In Definition 2.2, when comparing the defined type-I and type-II errors, I am confused about why in the type-II error, we need to assume $y$ is generated from the model $\hat{M}$, whereas in the type-I error, it is assumed that $y$ can be arbitrary. Wouldn't it make more sense to assume that $y$ is not generated from model $\hat{M}$ in the type-I error analysis? Based on the current definition, if $y$ is indeed generated by  $\hat{M}$, then the definition says its detection correctness is at most $\alpha$? Please clarify it.
>
>
> To clarify, our definition assumes an arbitrary *fixed* suspect text $y$. In particular, if $y$ is drawn from any distribution that is statistically independent to the (random) watermark key then we can condition on $y$ and the “no false positive” guarantee applies.  It covers any procedure (including human generation) that produces $y$ in a manner independent of the secret partition $G$. However, as the reviewer pointed out, it does not cover *random* text $y$ that depends on $G$. That would include the distribution from $\hat{M}$ and would not be reasonable.  The reason why we can handle an arbitrary fixed $y$ is because our Type I error analysis only depends on the distribution of $G$ — something we have full control over.
>
>
> ### **Extending Error Guarantees to General K-gram Method**
> > For the general K-gram method, can the error guarantees and analysis be extended? If so, how will this general K value place a role in the error analysis?
>
> Appendices D,E, and the Conclusion discussed results for K-gram for $k>1$. In short, our robustness guarantee, Type I error guarantee, and accuracy guarantee apply immediately with no changes in the analysis.  The Type II error guarantee can be extended to the case when $k >1$, but requires us to define the Martingale and bound the concentration differently. We will have a clearer discussion in the updated version.

---

> > ### Comment · Reviewer_Po1T · 2023-11-21
> >
> > I thank the authors for the response. I have no further comments.

---

### Official Review · Reviewer_L3yT · 2023-10-30

**Soundness:** 3 good
**Presentation:** 3 good
**Contribution:** 3 good
**Rating:** 6
**Confidence:** 3

**Summary:**

This paper is oriented towards addressing the misuse of Large Language Models (LLMs) in generating text, with a specific focus on the field of active detection, which offers advantages over passive detection. Firstly, a rigorous theoretical framework for LLM text watermarking is formally defined, considering the text quality, the correctness of detection, and robustness against post-processing. Subsequently, a specific construction is provided within the defined theoretical framework. This involves simplifying the grouping strategy of the vocabulary from the K-Gram watermark scheme of Kirchenbauer et al.[1], to fixed grouping in order to enhance robustness. The theoretical properties of the proposed watermark instance within the framework are then theoretically proven. Finally, experimental demonstrations of the evaluation metrics in the watermark framework are conducted across multiple datasets and language models.

Reference:
[1] John Kirchenbauer, Jonas Geiping, Yuxin Wen, Jonathan Katz, Ian Miers, and Tom Goldstein. A watermark for large language models. International Conference on Machine Learning, 2023.

**Strengths:**

Due to the low redundancy of the text carrier and the limited controllable information available during the gradual generation process, robustness becomes a challenge in the watermark task for active source tracing. This paper is focused on a specific 0-bit watermark task, namely human-machine text recognition, and provides a simple but robust approach to improve an existing method. The fixed grouping approach reduces the propagation of post-processing on text, which may result in some loss of text quality. However, experimental results indicate that it is comparable to the baseline. The key point is that it offers proof regarding the correctness and robustness of the proposed watermark.

**Weaknesses:**

This paper ensures the property of watermark quality through proof based on distribution distance. However, the relevant experiments only include perplexity and human assessment experiments, lacking distribution-based experimental validation. In addition, the definition of editing operations encompasses "paraphrase, insertion, deletion, replacement", but in the experimental section, robustness results for insertions are missing, which were replaced with "swap". What’s more, the experimental results about the impact of text length on watermark effectiveness are missing.

It may be beneficial to include the above experimental results.

**Questions:**

In the contribution summary section of the main text, the statement reads as follows: "To the best of our knowledge, we are the first to formulate the LLM watermarking as a cryptographic problem and to obtain provably robust guarantees for watermarks for LLMs against arbitrary edits." Could it be explained why the work of Christ et al.[1] is not considered "the first to formulate the LLM watermarking as a cryptographic problem"?

Reference:
[1] Miranda Christ, Sam Gunn, and Or Zamir. Undetectable watermarks for language models. arXiv preprint arXiv:2306.09194, 2023.

---

> ### Author Response · Authors · 2023-11-20
> **Rebuttal Response to Reviewer L3yT (Part 1/2)**
>
> Thanks for your valuable feedback! We address your concerns as below.
>
> ### **Lack of Distribution-Based Experimental Validation**
> > This paper ensures the property of watermark quality through proof based on distribution distance. However, the relevant experiments only include perplexity and human assessment experiments, lacking distribution-based experimental validation.
>
> We recognize the importance of conducting distribution-based experiments to validate our theoretical claims. In our study, we focus on the quality guarantee for the next token distribution, as described by the following equation:
> $
> \forall h, \max\left(D\_{KL}(\hat{p}\_t || p\_t)\, D\_{KL}(p\_t || \hat{p}\_t)\right) \leq \min{(\delta, \delta^2/8)}
> $
>
> To empirically validate this theory, we conducted experiments with various values of $\delta$ and calculated the average Kullback-Leibler (KL) divergence of the next word probability distribution for the generated text. The values of $\delta$ and the corresponding KL divergences are as follows:
>
> | $\delta$ | Average KL Divergence |
> |-|-|
> | 0.5 | 0.0199 |
> | 1.0 | 0.0877 |
> | 1.5 | 0.1698 |
> | 2.0 | 0.3179 |
> | 2.5 | 0.4749 |
>
> These results demonstrate that with an increase in $\delta$, there is a corresponding increase in the KL divergence, providing empirical support for our theoretical framework.
>
> ### **Robustness Results for Insertions**
> > In addition, the definition of editing operations encompasses "paraphrase, insertion, deletion, replacement", but in the experimental section, robustness results for insertions are missing, which were replaced with "swap".
>
> In addressing robustness against insertions, we opted to focus on "swap" operations. This decision was based on the understanding that, within a bounded edit distance, the effects of insertion and deletion operations are comparable. Specifically, each of these operations can at most transition one token from the green list to the red list. This analysis basis is detailed in Appendices C.2 and D.2.
>
> Further, we also conducted experiments involving random word insertions from a predefined list (randomly selected from the dictionary). The words selected for insertion included `["apple", "blue", "cat", "dog", "elephant", "flower", "giraffe", "house", "ice", "jelly", "kite", "lemon", "moon", "nest", "orange", "penguin", "quilt", "rose", "sun", "tree", "umbrella", "violet", "water", "xenon", "yacht", "zebra"]`, ensuring coverage of both green and red tokens. These words were randomly inserted at random positions within the text.
>
> The results, as summarized below, indicate that our method demonstrates superior detection capabilities compared to the baseline, even under increased interaction levels:
>
> | Insert Level | KGW+23 AUC | Our AUC |
> |-|-|-|
> | Insert-0.1  | 0.904 | 0.947 |
> | Insert-0.3  | 0.814 | 0.928 |
> | Insert-0.5  | 0.725 | 0.900 |
>
> Our approach consistently shows better detection results against the baseline across various interaction levels.
>
> ### **Impact of Text Length on Watermark Detection**
> >  What’s more, the experimental results about the impact of text length on watermark effectiveness are missing.
>
> To investigate the effect of text length on the effectiveness of our watermark detection method, we conducted experiments across different text lengths under both normal and attack scenarios. Specifically, we truncated the text to lengths of 50, 100, 150, and 200 tokens, and evaluated the detection performance. The attack is done by ChatGPT paraphrasing attack. The results, detailed below, demonstrate the robustness of our approach across varying text lengths.
>
> | Length | Setting | AUC   | TPR at 1% FPR | F1 score at 1% FPR | TPR at 10% FPR | F1 score at 10% FPR |
> |-|-|-|-|-|-|-|
> | 50     | Normal  | 0.993 | 0.892 | 0.932  | 0.987  | 0.899|
> | 100    | Normal  | 0.999 | 0.987         | 0.987              | 1.0            | 0.911               |
> | 150    | Normal  | 1.0   | 1.0           | 0.989              | 1.0            | 0.907               |
> | 200    | Normal  | 1.0   | 1.0           | 0.995              | 1.0            | 0.952               |
>
> | Length | Setting | AUC | TPR at 1% FPR | F1 score at 1% FPR | TPR at 10% FPR | F1 score at 10% FPR |
> |-|-|-|-|-|-|-|
> | 50     | Attack  | 0.886 | 0.297         | 0.451              | 0.696          | 0.731               |
> | 100    | Attack  | 0.948 | 0.525         | 0.683              | 0.858          | 0.835               |
> | 150    | Attack  | 0.968 | 0.661         | 0.787              | 0.915          | 0.863               |
> | 200    | Attack  | 0.985 | 0.829         | 0.896              | 0.953          | 0.883               |
>
> These results highlight the consistent and effective detection of watermarks across various text lengths, affirming the adaptability and robustness of our method. Generally, for shorter text, the detection is not perfect but still achieves very high AUC scores. As the length increases, the detection performance improves.

---

> > ### Author Response · Authors · 2023-11-20
> > **Rebuttal Response to Reviewer L3yT (Part 2/2)**
> >
> > ### **Comparison with the Work of Christ et al. (2023)**
> > > In the contribution summary section of the main text, the statement reads as follows: "To the best of our knowledge, we are the first to formulate the LLM watermarking as a cryptographic problem and to obtain provably robust guarantees for watermarks for LLMs against arbitrary edits." Could it be explained why the work of Christ et al.[1] is not considered "the first to formulate the LLM watermarking as a cryptographic problem"?
> >
> >
> > Our work is independent of the work of Christ et al. (2023), we actually have discussed their work on Page 2 and then again with more details (half a page) in Appendix A (titled "More on Related Work"). This section elucidates the nuances that set our work apart and elaborates on the specific aspects where our methodologies and theoretical frameworks diverge from those of Christ et al. The most prominent difference is that they require a strong “undetectability” guarantee but that made their watermark not as robust to edits and other evasion attacks.
> >
> > The “first to formulate it as a cryptographic problem” claim that the reviewer pinpointed was correct by the time we write the paper, but in light of the work of Christ et al., (2023)  and the older work by Aaronson (not public, but complete as of late 2022 according to the talk we cited) that were brought to our attention, we agree that it is no longer appropriate.  We will modify the statement into “To he best of our knowledge, we are the first to provide a complete theory for the Kirchenbauer-style statistical watermarks and the first to obtain provable robustness guarantees for LLM watermarks against arbitrary edits.”

---

### Official Review · Reviewer_ivTE · 2023-11-01

**Soundness:** 3 good
**Presentation:** 3 good
**Contribution:** 3 good
**Rating:** 6
**Confidence:** 2

**Summary:**

The paper proposes a novel watermarking schema for text generators. They claim the proposed watermarking method possesses several nice properties that increase the feasibility of their approach, guaranteed generation quality, correctness in watermark detection and robustness against text editing.

**Strengths:**

+ The research direction is intriguing and timely.


+ The design of the $\delta$ increase for tokens in the red list is simple and intuitively correct, although the proof is very complex and I cannot go through the details in the proof.

+ One of the properties (robust to paraphrasing) is essential to watermarks, although this property is mainly proved empirically not supported by theoretical discussion.

**Weaknesses:**

Some points are overclaimed, e.g., contribution 1 claims the framework is theoretically rigorous. Some related work is not properly involved in discussion and comparison. All these points need careful consideration.

+ The robustness to edits within a bound is an awesome property and can be theoretically proved, but the robustness to paraphrasing models could hardly be proven, as there is no guarantee in the *edit distance* of paraphrasing. The related claims should be treated carefully.

+ The consistency of the output word distribution is a great property and it was also discussed in previous literature, such as [1]. How will you compare this approach with theirs? The Type I/II errors were not discussed in their work, can you show your advantages against them?

+ The robustness claim on paraphrase should be constrained by empirical findings and the paraphraser should not be aware of the watermarking method as they can still conduct attacks on this method. One extreme example could be totally random choice of synonyms and another interesting paraphrasing method is to use back-translation as paraphrasing with changes of style [2].

[1] He, Xuanli, et al. "Cater: Intellectual property protection on text generation apis via conditional watermarks." Advances in Neural Information Processing Systems 35 (2022): 5431-5445.

[2] Prabhumoye, Shrimai, et al. "Style Transfer Through Back-Translation." Proceedings of the 56th Annual Meeting of the Association for Computational Linguistics (Volume 1: Long Papers). 2018.

**Questions:**

Can you confirm the main difference between Algorithm 1 in this work and Algorithm 2 in [3]? IMO, this is an essential question to understand the main technical contribution of this paper.

It would be nice to link the 'confidence' of the watermarked outputs with the length of a sentence. Usually, longer sentences provide more evidence for watermarking.

---
**Minor:**

The examples in Algorithm 2 can be placed in comments.

[3] A Watermark for Large Language Models

---

> ### Author Response · Authors · 2023-11-20
> **Rebuttal Response to Reviewer ivTE (Part 1/2)**
>
> Thank you for your insightful review. We appreciate the opportunity to clarify and expand upon key aspects of our work.
>
> ### **Response to Paraphrasing Attacks**
> > The robustness to edits within a bound is an awesome property and can be theoretically proved, but the robustness to paraphrasing models could hardly be proven, as there is no guarantee in the edit distance of paraphrasing. The related claims should be treated carefully.
>
>
> Yes, our theoretical guarantee of robustness concerns edits, as discussed in the paragraph preceding Section 2.2. We acknowledged, “Admittedly, there are other attacks where edit distance does not capture either the effort or the utility loss. For example, if one prompts an unwatermarked LLM to paraphrase $y$ then the number of edits can be large but the semantic meaning is retained. However, edit distance is a natural metric that smoothly interpolates the gray zone between the world where $y_A = y$ in which it should clearly be caught and the other world where $y_A$ is independently created without using the watermarked model in which it would be a false positive if Detect returns 1.”
>
> Moreover, our theoretical framework does offer insights regarding the edit distance in paraphrasing. Paraphrasing can be seen as a sequence of actions – deletion, insertion, and replacement of tokens. Although extensive paraphrasing might result in a large edit distance (with the maximum being n, the length of the text), our framework assures that we can detect paraphrases if the edit distance falls below a certain threshold.
>
> Empirically, we found our watermark can still detect paraphrased text, but the “power” gets smaller, which means it requires longer watermarked text to have high confidence in detection. Our hypothesis is that paraphrasing reduces the density of watermarks in the text, but do not remove it.
> Overall, while paraphrasing attacks present challenges, edit distance offers a useful starting point. Further research on formal definitions and robustness against paraphrasing is important for future work.
>
> ###  **Output Word Distribution and Related Work**
> > The consistency of the output word distribution is a great property and it was also discussed in previous literature, such as [1]. How will you compare this approach with theirs? The Type I/II errors were not discussed in their work, can you show your advantages against them?
>
> We appreciate the reference to He et al. (2022) and will include a discussion in our revised paper.
> However, there are several key differences between these two works:
>
> (1) We are studying different tasks. Our paper addresses the problem of detecting AI-generated text, whereas CATER (He et al., 2022) focuses on protecting against model distillation. CATER involves watermarking training data so that models trained on it produce outputs containing the watermark. Its goal is to identify model/data theft, not detect AI-generated text. As such, CATER does not consider the Type I/II errors relevant to AI detection systems.
>
> (2) CATER uses post-processing to insert watermarks by changing words to synonyms. Our method integrates watermarking into the text generation process itself.
>
> (3) CATER addresses the output word distribution to prevent attackers from identifying watermarked words through statistical analysis. Our method provides a theoretical guarantee that the original and watermarked probability vectors for the next token are closely aligned in any Renyi-divergence. However, we do not guarantee the output word distribution is identical. We address the potential for attacks in Appendix B.5, where we explore the possibility of adversaries bypassing detection by using secret tokens estimated from the output word distribution. We find that it is difficult to accurately estimate the green list utilizing the output word distribution. Even if the green list is known, our watermark is still somewhat effective thanks to our added robustness.

---

> > ### Author Response · Authors · 2023-11-20
> > **Rebuttal Response to Reviewer ivTE (Part 2/2)**
> >
> > ### **Robustness to Paraphrasing Models**
> > > The robustness claim on paraphrase should be constrained by empirical findings and the paraphraser should not be aware of the watermarking method as they can still conduct attacks on this method. One extreme example could be totally random choice of synonyms and another interesting paraphrasing method is to use back-translation as paraphrasing with changes of style [2].
> >
> > We have conducted additional experiments to evaluate the robustness of our method to paraphrasing models. As discussed in Appendix B.5, we simulated a stronger white-box evasion attack. In situations where the adversary has either an estimated version or full knowledge of the green and red lists, they can formulate an evasion strategy. We simulate this by assuming the adversary employs WordNet from NLTK to identify token synonyms. Tokens identified as in the green list are replaced with red list synonyms, noting that some tokens may not have synonyms or may only have green synonyms. The results in Table 6 show it is difficult to evade detection even with known green list tokens. The detection AUC for the watermarked text is still somewhat high. In addition, the honest attempt to evade the attack by automatic synonym replacement has led to a significant drop in the text quality.
> >
> > During the rebuttal phase, we expanded our tests to include back-translation as a form of paraphrasing attack. Using a model from [1], we found that the back-translation attack was less effective compared to other methods like ChatGPT or DIPPER paraphrasing. Our results on the OpenGen dataset were as follows:
> >
> > **Our method**
> >
> > - AUC: 0.999
> >
> > - TPR at 1% FPR: 0.996, F1 score at 1% FPR: 0.985
> >
> > - TPR at 10% FPR: 1.0, F1 score at 10% FPR: 0.893
> >
> >
> >
> > **Baseline (Kirchenbauer et al., 2023)**
> >
> > - AUC: 0.997
> >
> > - TPR at 1% FPR: 0.942, F1 score at 1% FPR: 0.956
> >
> > - TPR at 10% FPR: 0.996, F1 score at 10% FPR: 0.886
> >
> > Both our method and the baseline demonstrated robustness against the back-translation attack, with our method exhibiting superior robustness compared to the baseline.
> >
> >
> > ### **Distinction Between Our Algorithm and the Baseline (Soft-Watermark) Method**
> > > Can you confirm the main difference between Algorithm 1 in this work and Algorithm 2 in [3]? IMO, this is an essential question to understand the main technical contribution of this paper.
> >
> >
> > The key difference is that we are using a fixed Green-Red split consistently and the soft watermarking (Kirchenbauer et al., 2023) method divides the vocabulary into a “green list” and a “red list” based on the prefix token. We discussed this in the related work and Appendix D (Analysis of Kirchenbauer et al., 2023).
> >
> > ### **Correlation Between 'Confidence' of Watermarked Outputs and Sentence Length**
> > > It would be nice to link the 'confidence' of the watermarked outputs with the length of a sentence. Usually, longer sentences provide more evidence for watermarking.
> >
> >
> > Regarding the correlation of watermark 'confidence' with sentence length, our designed z-score metric is indeed pertinent. We observe that the z-score increases with the length of the text, which typically corresponds to a smaller P-value. This relationship underscores the reliability of our watermarking approach, particularly in longer text segments, enhancing the robustness of our detection methodology.
> >
> > ------
> > [1] Edward Ma. NLP Augmentation. 2019. https://nlpaug.readthedocs.io/en/latest/augmenter/word/back_translation.html

---

> > > ### Comment · Reviewer_ivTE · 2023-12-04
> > > **Acknowledgement**
> > >
> > > Thank you for your responses. I acknowledge they are taken into consideration.

---

### Meta-Review · Area_Chair_RW7z · 2023-12-04

**Metareview:**

The paper introduces a novel watermarking scheme targeting text generators, emphasizing properties like robustness against text editing, guaranteed generation quality, and correctness in watermark detection. Reviewers acknowledge the intriguing research direction and the paper's solid theoretical foundation. Concerns are raised regarding overclaiming, lacking certain experimental validations such as distribution-based assessments and impact analysis of text length on watermark effectiveness. The rebuttals clarified most of the points and addressed most of the concerns. Reviewer's questions on false positives and comparisons with the K-gram method were addressed satisfactorily. I would encourage the reviewers to supplement experiments to assess performance on shorter texts. Despite these concerns, considering the significant contributions and the paper's overall solidity, the recommendation is acceptance.

**Justification For Why Not Higher Score:**

Despite the paper's strengths and valuable contributions, several issues regarding lacking some experiments and overclaiming warrant caution in assigning a higher score. Addressing these concerns through additional experiments, clearer comparisons, and more nuanced claims could significantly strengthen the paper.

**Justification For Why Not Lower Score:**

While there are indeed areas for improvement, the positive aspects, such as the paper's theoretical soundness and empirical demonstrations of improved performance, justify not assigning a lower score. These positive elements contribute significantly to the paper's overall merit and make it worthy of consideration for acceptance, even with acknowledged areas for enhancement.

---

### Decision · Program_Chairs · 2024-01-16

Accept (poster)